# Chemotherapy-driven intestinal dysbiosis and indole-3-propionic acid rewire myelopoiesis to promote a metastasis-refractory state

Ludivine Bersier [1], L. Francisco Lorenzo-Martin [2], Yi-Hsuan Chiang[1], Stephan Durot [3], Aleksander Czauderna[4,5], Tural Yarahmadov[6,7], Tania Wyss Lozano [1,8], Irena Roci [1], Jaeryung Kim [1,16], Nicola Zamboni [3], Nicola Vannini [1], Caroline Pot [9], Tinh-Hai Collet [10,11], Deborah Stroka [6,7], Jeremiah Bernier-Latmani [1], Matthias P. Lutolf [2,12], Simone Becattini[4,5], Thibaud Koessler [13,14] & Tatiana V. Petrova [1,15] ✉

The contribution of chemotherapy-induced tissue injury to individual susceptibility to metastasis remains largely unexplored. We report that chemotherapy indirectly prevents colorectal cancer (CRC) liver metastases by inducing a lasting systemic "chemomemory". Chemotherapy-induced intestinal mucositis alters nutrient availability, promoting the expansion of tryptophan-metabolizing bacteria and production of the microbial metabolite indole-3-propionic acid (IPA). IPA reprograms bone marrow myelopoiesis by redirecting common myeloid progenitor fate toward the macrophage lineage, limiting generation of immunosuppressive Ly6C$^{high}$CCR2$^+$ monocytes. This shift enhances CD4$^+$ T cell antitumor function by promoting Th1 differentiation and spatially reorganizing CD8$^+$ and CD4$^+$ T cell interactions within the metastatic microenvironment. In a subset of CRC patients, circulating IPA levels increase after chemotherapy and inversely correlate with monocyte abundance, while high monocyte levels were associated with reduced survival. Our findings reveal that chemotherapy-induced intestinal injury normalizes pathological myelopoiesis through a microbiota-derived metabolite and identify IPA as a potential adjuvant to counteract monocyte-driven immunosuppression and metastasis.

Colorectal cancer (CRC) is the world-wide second leading cause of cancer-related mortality[1]. Stage I and II CRC tumors can be surgically resected and are associated with an excellent prognosis. Recently, spectacular responses were reported for microsatellite instable CRCs treated with immune-check point inhibitors such as anti-PD-1[2]. However, this cancer subtype represents only 5-13% of all CRC cases, highlighting the need to improve therapy for the remaining 85-95% patients with microsatellite stable (MSS) CRCs. Fluoropyrimidine-based chemotherapy in combination with oxaliplatin after curative surgery remains a mainstay treatment for selected high-risk stage II and stage III CRCs. It minimizes tumor recurrence and prolongs survival[3], however, a significant proportion of patients (30-50%) develop distant recurrence, mainly in the liver[4].

The importance of microbiota for acute cytotoxic effect of chemotherapy has been demonstrated. Absence of MYD88 microbial product-sensing in the myeloid compartment of germ-free mice reduces oxaliplatin treatment effectiveness[5]. In addition, cyclophosphamide-induced bacteria translocation to secondary lymphoid organs stimulates an antitumor CD4+ T cell response, essential for optimal cyclophosphamide efficacy[6]. In the adjuvant settings, patients receive perioperative antibiotics. However, how the microbiota influence metastatic relapse in the adjuvant setting has not been addressed yet. A recent large retrospective clinical study EVADER-1 demonstrated that receiving antibiotics prior or during chemotherapy treatment significantly reduced disease-free survival in CRC patients[7], suggesting a contribution of the microbiota or microbial-derived products. Yet, the mechanistic link between chemotherapy, microbiota alterations and cancer progression, including liver metastasis development, remains incompletely understood.

Here, we provide a part of the mechanistic explanation for this clinical observation. We demonstrate that prior exposure to standard of care 5-FU and oxaliplatin chemotherapy establishes a liver metastasis refractory state by durably rewiring gut microbiota to produce the tryptophan-derived metabolite, indole-3-propionic acid (IPA). We further identify myeloid cells as primary target of IPA, contrasting with previous observations[8]. IPA supplementation biased common myeloid progenitors fate toward the macrophage lineage, reducing immunosuppressive Ly6C$^{high}$CCR2+ monocyte output with minimal direct impact on lymphoid cells or T cell function. The reduction of immunosuppressive myeloid cells allowed expansion and activation of anti-tumor CD4+ T cells which limited CRC metastasis formation. The effects observed in mouse models are clinically relevant, as circulating IPA levels are negatively correlated with abundance of monocyte and are increased post-chemotherapy in a subset of chemo-naïve CRC patients. Our study underscores the necessity of characterizing the systemic effects of chemotherapy to anticipate the outcomes of cancer treatment and to discover new therapeutic strategies.

## Results

### 5-FU and oxaliplatin exposure provide indirect lasting protection against CRC liver metastasis

Up to 50% of CRC patients develop intestinal mucositis after the first round of chemotherapy[9], which may trigger the local and systemic release of cytokines, growth factors, and bacterial-derived products influencing subsequent metastatic relapse. To determine how chemotherapy induced intestinal mucositis influences metastatic relapse in the context of adjuvant CRC treatment, we first tested if a clinically relevant chemotherapeutic regimen (FO) consisting of one dose of 3.5 mg/kg oxaliplatin and one hour later 50 mg/kg 5-FU, followed by 50 mg/kg 5-FU on the next day directly inhibited CRC growth and induced intestinal mucositis (Fig. 1a)[10]. The FO-treated group displayed moderate, but significant, weight loss and a tendency towards reduced intestinal length 3 days after treatment initiation (Supplementary Fig. 1a, b). Analysis of the small intestine 3 days post-FO revealed reduced crypt cell proliferation, villi length and crypt depth (Fig. 1b, c), consistent with a cytotoxic effect of FO on transient amplifying cells. To assess both mucositis induction and tumor response, the FO regimen was initiated two weeks after implantation of Apc$^{fl/fl}$; Kras$^{LSLG12D}$; Tp53$^{fl/fl}$; Vil-CreERT2 (AKP) tumor cells (Fig. 1d)[11]. Tumor weights were comparable between control and FO-treated mice five days post-treatment (Supplementary Fig. 1c). However, we observed decreased tumor cell proliferation and increased DNA double-strand breaks in FO-treated tumors, confirming the antitumor activity of FO (Supplementary Fig. 1d–g). We observed increased crypt depth and transit-amplifying cell proliferation in small intestine, indicative of ongoing intestinal regeneration five days post-FO (Fig. 1e, f). In contrast, the colon crypt proliferation and depth were similar between control and

treated mice at the same time point (Supplementary Fig. 1h, i), aligning with human data showing oxaliplatin preferentially induces ileal cell death and spares healthy colon tissue in patients[12]. Accordingly, expression of proinflammatory cytokines Tnf (encoding TNFα) and Il1b was increased in the ileum at day 5 post-treatment (Supplementary Fig. 1j). Analysis of immune response revealed increased infiltration of myeloperoxidase (MPO+) neutrophils in the FO-treated ileal lamina propria at day 3 and 5 post-FO, while CD4+ and CD8+ T lymphocyte levels were unaffected (Supplementary Fig. 1k–r). These results confirm that the clinically relevant FO regimen directly inhibits CRC tumor growth and causes intestinal mucositis.

The liver is the most common site of distant spread for CRC, with liver metastases developing in up to 25% of patients undergoing surgery and adjuvant chemotherapy. FO has a relatively short half-life and is eliminated 4 days following acute administration[13,14]. To determine if FO chemotherapy-induced intestinal injury affects liver metastasis burden independently of its direct cytotoxic effect and to mimic the adjuvant treatment, we intrasplenically injected mice with either Apc$^{-/-}$; Kras$^{LSLG12D}$; Tp53$^{-/-}$;Smad4$^{-/-}$-mCherry (AKPS) organoids or MC38-GFP mouse colon adenocarcinoma cells five days after FO treatment (Fig. 1g). In this experimental setup, liver metastases form after the active drug has been cleared (Fig. 1g). AKPS and MC38 cell lines are models of microsatellite stable and instable CRC, respectively, and allowed testing if CRC subtype alters responses to FO pre-conditioning[15,16]. FO pre-conditioning markedly reduced liver metastases in both the AKPS and MC38 models (Fig. 1h–k), indicating a lasting anti-metastatic effect of chemotherapy, independent of its direct impact on tumor cells. We next investigated the duration of FO pre-conditioning-mediated tumor growth inhibition (Fig. 1l). Injection of MC38-GFP tumor cells ten days after FO pre-conditioning reduced metastasis formation, indicating that the metastasis refractory state is lasting (Fig. 1l–n; Supplementary Fig. 2a, b). Injecting MC38-GFP tumor cells twenty days after FO pre-conditioning, mimicking the end of a treatment cycle in patients, showed a tendency for persistence of metastasis refractory state (Supplementary Fig. 2c–e). In all, we demonstrate that FO-induced mucositis and its systemic effects maintain a lasting anti-tumor effect.

To determine if decreased liver metastatic burden was due to hepatotoxicity preventing metastatic cell implantation, we analyzed hepatic transaminase levels in the control and FO-treated mice. However, alanine aminotransferase (ALAT) and aspartate aminotransferase (ASAT) levels were similar between control and FO-treated mice indicating that direct hepatotoxicity is not inhibiting metastatic growth (Supplementary Fig. 2f, g). During metastasis, disseminated tumor cells extravasate to distant tissues within 24 h[17]. To test whether FO pre-conditioning inhibited survival of cancer cells in circulation or reduced their extravasation capacity, we analyzed the presence of MC38-GFP cells in the liver 24 h after their injection into the spleen by FACS (Supplementary Fig. 2h). The number of MC38-GFP cells was similar between FO pre-treatment and control mice, suggesting FO impacts metastatic outgrowth rather than earlier steps of metastatic dissemination (Supplementary Fig. 2i, j). In summary, these results indicate that pre-exposure to the standard-of-care CRC chemotherapy generates a durable metastasis-refractory state which we term "chemomemory" (Fig. 1o).

### Anti-metastatic chemomemory is microbiota-dependent

Chemotherapeutic drugs not only impact cancer cells, but also the tumor microenvironment and gut microbiota composition and functions[12,18,19]. To assess if gut microbes are necessary for chemomemory, we depleted the microbiota using broad-spectrum antibiotics and assessed liver metastatic burden after FO pre-conditioning (Fig. 2a)[20]. Analysis of fecal 16S rRNA gene abundance confirmed

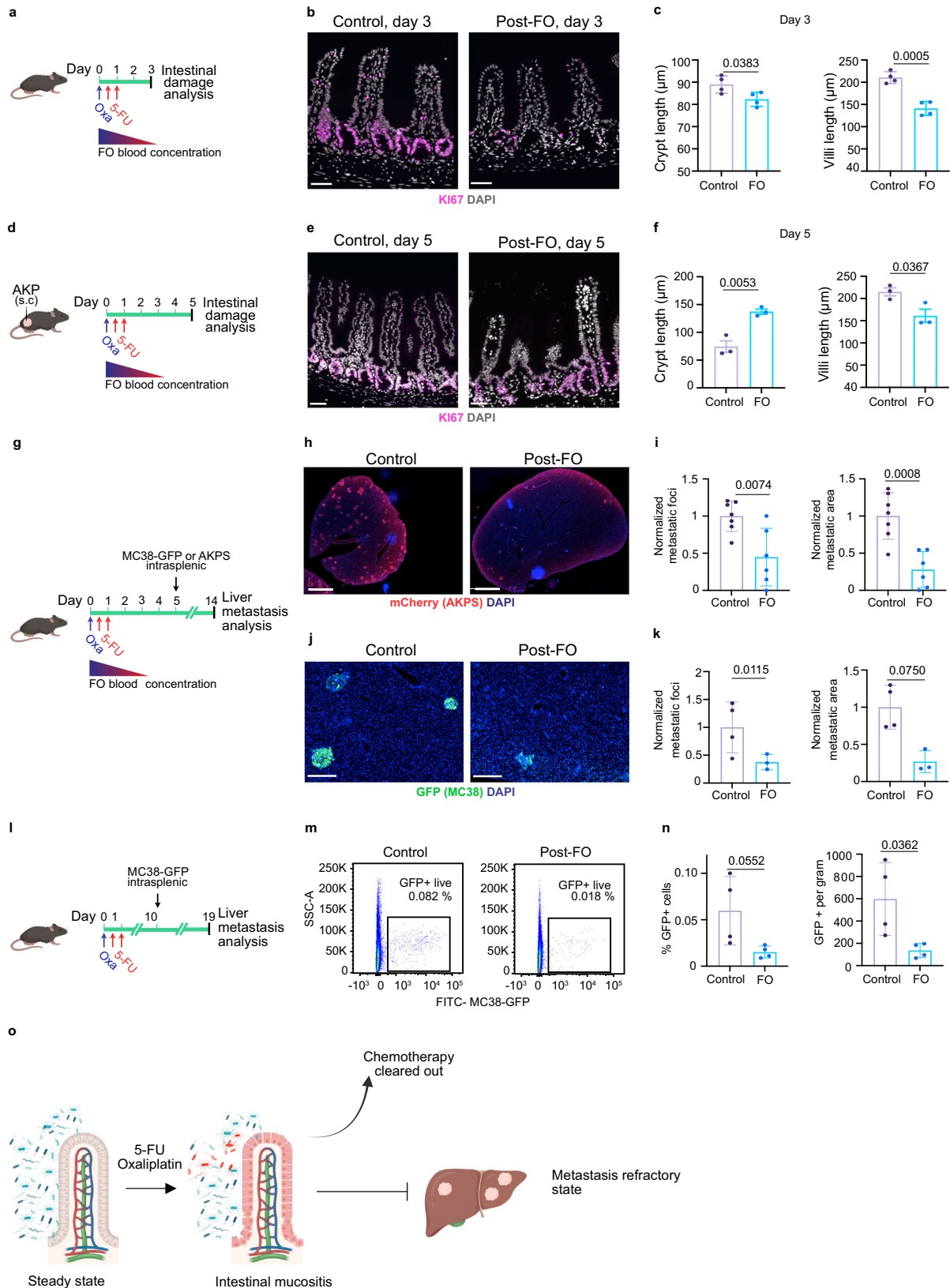

efficient microbiota depletion in the antibiotic-treated group (Supplementary Fig. 3a). Liver metastases grew significantly more in antibiotic-treated mice compared to controls, despite FO pretreatment (Fig. 2b–d). Fecal bacterial load negatively correlated with metastatic load (Fig. 2e), further suggesting an important role for the gut microbiota in promoting chemomemory.

## FO-induced gut damage increases gut microbe diversity by altering nutrient availability

To determine how chemotherapy modulates the intestinal microbiota composition, we performed 16S rRNA sequencing of mouse feces 5 days post-FO treatment. Higher alpha and beta diversity have been associated with better responses to chemotherapy in breast cancer

**Fig. 1 | Prior exposure to FOLFOX adjuvant therapy generates liver metastasis refractory state. a** Scheme of the experiment. Wild-type mice received 5-FU +oxaliplatin (FO) or PBS (control); intestines were harvested 3 days later. **b** FO reduced proliferation of intestinal crypt cells. KI67 (magenta) and DAPI (grey). Scale bars: 100 μm. **c** Quantification of crypt (left) and villi (right) length 3 days post-FO. $n = 4$ per condition. **d** Scheme of the experiment. AKP organoids were injected subcutaneously and FO treatment was initiated when tumor reached 130 mm³. Tissues were harvested 5 days after FO. **e** Hyperproliferative crypt cells five days after FO. KI67 (magenta) and DAPI (grey). Scale bars: 100 μm. **f** Quantification of crypt (left) and villi (right) length five days post-FO. $n = 3$ per condition. **g** Scheme of the experiment. AKPS-mCherry or MC38-GFP cells were injected intrasplenically 5 days after FO preconditioning, when active form of FO has been cleared-out. **h** FO preconditioning reduces liver metastasis formation in AKPS model. Staining for tumor cells (mCherry, red) and DNA (blue). Scale bar: 2 mm. **i** Quantification of

AKPS metastasis area and number per liver area normalized to control mean. Data are from two independent experiments. Control, $n = 7$. FO, $n = 6$. **j** FO pre-conditioning reduces liver metastasis formation in MC38-GFP model. Staining for GFP (green) and DNA (blue). Scale bar: 200 μm. **k** Quantification of MC38-GFP metastasis area and number per liver area normalized to control mean. Control, $n = 4$. FO, $n = 3$. **l** Scheme of the experiment. MC38-GFP cells were injected intrasplenically 10 days after FO preconditioning. **m** FO chemomemory is lasting. Representative FACS plots of GFP+ cancer cells in livers of control and FO preconditioned mice. **n** Quantification of percentage and count of GFP positive cancer cells in the indicated conditions. $n = 4$ per condition. **o** Chemotherapy exposure exerts a lasting antimetastatic effect on CRC liver metastasis. Data are shown as mean ± SD and analyzed using two-tailed unpaired Student's t- test (**c**, **f**, **i**, **k**, **n**). Icons were created in *BioRender under the license SABINE, A. (2025)* https://BioRender.com/ pmy8rs7. Source data are provided as a Source Data file.

patients and there was greater alpha diversity in FO-treated intestines compared to PBS-treated controls (Supplementary Fig. 3b)[21,22]. There were also major changes in the bacterial community composition with clear separation between the two treatment groups (Supplementary Fig. 3c). FO promoted significant expansion of Firmicutes and Proteobacteria at the phylum level and the class Clostridia including Oscillospirales, Christensenellales, Lachnospirales at the order level (Fig. 2f, g). Selective expansion of these two bacterial phyla is consistent with reports that some Firmicutes are resistant to 5-FU cytotoxicity, while intestinal inflammation increases Proteobacteria abundance[23,24].

To characterize how FO affects the small intestine metabolome, we performed metabolomic analysis of the ileum 5 days post-FO. Ileum from FO-treated mice displayed an increase in 53 metabolites and a decrease in 18 metabolites (Fig. 2h, Supplementary Data 1). Most notably, we observed increased abundance of compounds associated with the pro-inflammatory arachidonic acid metabolism pathway (Fig. 2h, Supplementary Data 1)[25]. In addition, FO-treated ileum contained less citrulline (Fig. 2h, Supplementary Data 1), a metabolite mostly produced by enterocytes and used as a marker of enterocyte mass[26]. Collectively, together with histological and immune infiltration analyses, these changes indicate that FO treatment transiently decreases enterocyte absorptive capacity and promotes intestinal inflammation.

FO-induced intestinal mucositis significantly decreased small intestinal villus size and hence the absorptive surface (Fig. 1b, c). Gut-resident microbes rely on specific nutrient sources[27–29], therefore we hypothesized that altered colonic microbiota composition post-FO was due to differences in gut lumen nutrient availability. Consequently, we performed untargeted metabolomics on cecal contents from FO and control-treated mice. Over Representation Analysis (ORA) of metabolite subclass enrichment revealed that amino acids, peptides, and amino-acid metabolism by-products were the most significantly increased metabolites in FO-treated mice caeca (Fig. 2i)[30]. Amino acids and peptides are absorbed in the proximal small intestine[31]. Therefore, the observed higher cecal levels of single, and particularly dipeptide, amino acid content (Fig. 2i, j; Supplementary Data 2) is due to a reduction in the absorption capacity caused by small intestinal mucositis. Of interest, proteins are the preferred carbon and nitrogen source for Firmicutes in vivo, suggesting increased peptide concentration drives expansion of these taxa[29]. In all these data indicate that FO-induced intestinal mucositis modifies nutrient availability to the gut microbiota, which increases the abundance of Firmicutes and in particular the Clostridia class bacteria.

## FO enhances circulating levels of microbial-derived indole-3-propionic acid

Since the anti-metastatic effects of FO pre-conditioning were gut microbiota-dependent and FO increased the abundance of

Clostridia, we hypothesized that these bacterial populations produce a soluble factor that mediated chemomemory. Therefore, we performed untargeted polar metabolomics of portal blood 5 days post-FO (Fig. 3a) and found 95 and 68 metabolites were significantly increased or decreased by FO, respectively (Fig. 3b; Supplementary Data 1). Indole-3-propionic acid (IPA) was consistently increased in portal blood following FO treatment along with bile acid related metabolites and amino acids (Fig. 3c). Retention time comparison with IPA pure standard confirmed the IPA identity detected by untargeted flow injection analysis (Supplementary Fig. 4a).

Analysis of functional microbiome-encoded pathways using PICRUSt2 revealed an enrichment in tryptophan metabolism among taxa increased after FO treatment (Fig. 3d)[32]. Importantly, IPA is a by-product of tryptophan metabolism and produced exclusively by Clostridia and in particular *Clostridiaceae*, a genus increased upon FO treatment (Supplementary Fig. 4b)[33]. The initial targeted metabolomics analysis showed an increase of IPA concentration in FO-treated serum over the upper detection limit (Supplementary Fig. 4c). Consequently, we performed the targeted LC-MS/MS in a second facility for the samples reaching detection limit and this second analysis further validated a > 10-fold IPA increase in the majority of FO-treated mice (Fig. 3e).

To determine if the increase in IPA after FO was due to altered microbial metabolism, we analyzed metabolites in portal blood of mice co-treated with FO and antibiotics or corresponding controls. Antibiotic treatment significantly modified the portal metabolomes of control and FO-treated mice (Fig. 3f; Supplementary Fig. 4d; Supplementary Data 3). As expected, microbiota depletion led to accumulation of taurocholic acid, a primary bile acid, and a reduction in the abundance of microbial derived products such as secondary bile acid, 12-ketodeoxycholic acid, hippurate and *p*-cresol sulfate (Fig. 3f; Supplementary Data 3)[34]. The above metabolites were not modified upon FO treatment. In contrast, IPA levels were signficantly increased after FO treatment in a microbiota-dependent manner (Fig. 3f, g). Furthermore, IPA abundance positively correlated with fecal bacterial load, as determined by 16S rRNA analysis (Supplementary Fig. 4e). In comparison, FO-dependent increases in proline, oxoadipate and cholate were not significantly modified by microbiota depletion (Supplementary Fig. 4f).

To assess if FO-induced increased levels of circulating IPA were durable, we measured polar metabolites from portal serum sampled 10 days after FO treatment. Among the metabolites increased at day 5, IPA and oxoadipate were still increased at day 10 (Supplementary Fig. 4g). Taken together, these data indicate that FO treatment increases the abundance of gut microbes which preferentially produce IPA, and this metabolite persists in portal blood long after FO is removed from the system (Fig. 3h).

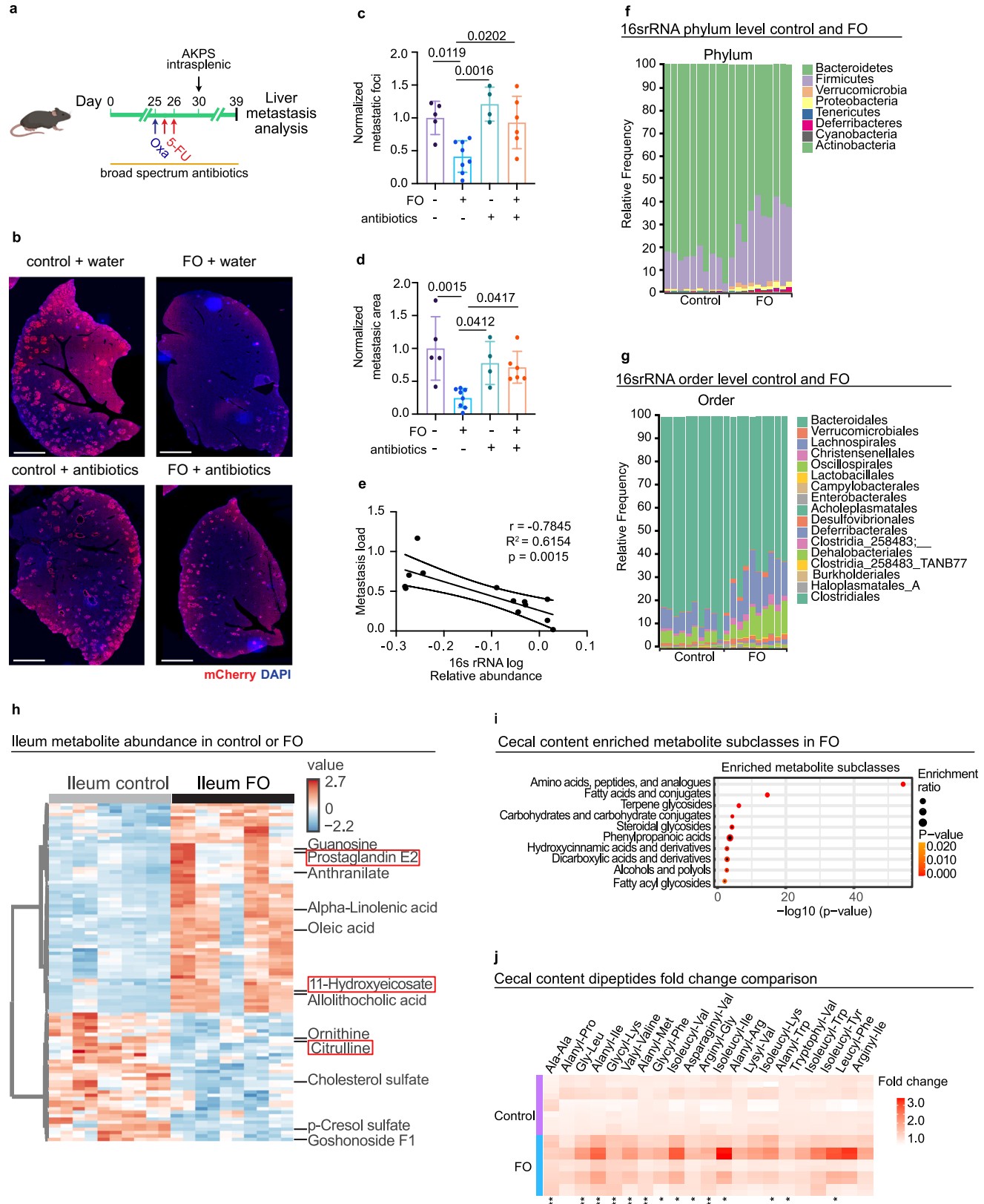

### Exogenous IPA recapitulates FO chemomemory by modulating the immune system

Since blood IPA concentration was increased after FO in a microbiota-dependent manner (Fig. 3c, g), we hypothesized that IPA was functionally important for FO-mediated chemomemory. Therefore, we first tested the ability of oral IPA supplementation to limit growth of AKPS and MC38 liver metastases (Fig. 4A). We chose an IPA dose of 20 mg/kg previously shown to promote physiologically relevant circulating IPA concentrations in mouse models[35]. IPA supplementation significantly reduced CRC liver metastatic load when given either before and/or after metastasis establishment (Fig. 4A–E; Supplementary Fig. 5a–c) and reduced cancer cell proliferation (Fig. 4F, G). We next tested if IPA

**Fig. 2 | Chemomemory anti-metastatic effect is microbiota-dependent.**
**a** Scheme of the experiment. Mice were treated with broad-spectrum antibiotics for three weeks before FO treatment and AKPS were injected intrasplenically five days after FO or vehicle treatment. **b** Microbiota depletion abolishes chemomemory. Staining of liver colonized by AKPS tumor cells for mCherry (red) and DNA (blue). Scale bar: 2 mm. **c**, **d** Quantification of metastatic area and number of metastases per liver area normalized to control mean. Data are from two independent experiments. Control + water, $n = 5$. FO + water, $n = 8$. Control + antibiotics, $n = 4$. FO + antibiotics, $n = 6$. **e** Anti-correlation between liver metastasis load and gut bacterial content. qPCR for 16Sr RNA. FO, $n = 8$. FO+antibiotics, $n = 6$. Taxonomic relative frequency of bacterial composition at phylum (**f**) and order (**g**) level from stool analyzed 5 days post FO or PBS, reveals expansion of Firmicutes and Clostridia class. $n = 10$ per condition. Data from two independent experiments. **h** Hierarchical clustering heatmap showing metabolite abundance changes in control and FO treated ileum (logFc2 > 0.2; FDR < 0.05). $n = 5$ per condition run in duplicate. **i** Accumulation of amino acids and peptide-related products in the cecum of FO-treated mice. Metabolites subclass enrichment analysis was run on the upregulated metabolites in cecal content of mice treated with FO or PBS. log2fc > 0.2; adjusted p-value < 0.1. $n = 5$ per condition. **j** Heatmap of relative abundance of dipeptides in cecal content of control and FO-treated mice. $n = 5$ per condition. *$P < 0.05$, **$P < 0.001$ and ***$P < 0.0001$. Data are shown as mean ± SD and analyzed using Student's t-test and corrected for multiple hypothesis testing with the BH method (**h**, **j**), one-way Anova with Tukey's multiple comparisons test (**c**, **d**) and Pearson correlation test (**e**). Icons were created in *BioRender under the license SABINE, A. (2025)* https://BioRender.com/ pmy8rs7. Source data are provided as a Source Data file.

could re-establish chemomemory in the absence of gut microbiota (Fig. 4H). Supplementing IPA to microbiota-depleted mice carrying intrasplenically injected AKPS cells reduced liver metastatic burden to levels similar to those observed in FO-treated, microbiota-competent mice (Figs. 4I, J and 1h, i). In all, these results show that dietary IPA supplementation is sufficient to recapitulate FO-induced anti-metastatic chemomemory.

To test if IPA directly inhibits cancer cell growth, we treated CRC tumoroids from patients and mouse AKPS organoids with IPA in vitro and analyzed cancer spheroid cell growth. The concentration of IPA in human blood ranges from 1 to 10 μM[36–38], while in mice functionally relevant in vivo concentrations of IPA range from 3 to 15 μM[35,39]. Therefore, we tested the effects of high physiological (10 μM) and supraphysiological (50, 100 and 200 μM) IPA concentrations[36,37]. Only non-physiological concentration of 100 μM IPA inhibited growth of patient-derived MSS tumoroids (Supplementary Fig. 5d, e), while mouse organoids were resistant (Supplementary Fig. 5f, g). These results suggest that the anti-metastatic effect of IPA in vivo is primarily mediated by IPA's indirect effects on the tumor or organ microenvironment. To test the contribution of immune cells, we analyzed hepatic AKPS metastasis growth after IPA administration in NOD-*scid Il2rg*[null] (NSG) mice lacking B, T and functional NK cells (Fig. 4K). IPA was unable to suppress metastatic burden or tumor cell proliferation in NSG mice (Fig. 4L–O). These results indicated IPA anti-metastatic activity is at least in part mediated by its effect on immune cells.

A recent study reported that IPA enhanced antitumor immunity by directly inducing TCF1 expression in CD8[+] T cells thereby promoting the expansion of CD8[+]TCF1[+] stem-like cells[8]. To test if IPA is directly affecting T cell activation, we exposed CD3e and CD28 antibody-activated CD4[+] or CD8[+] T cells to physiological (10 μM) and supraphysiological (50, 500 μM) concentrations of IPA in vitro (Supplementary Fig. 5h, i). However, IPA treatment did not impact T cell proliferation nor activation in vitro (Supplementary Fig. 5j, k). CD4[+] and CD8[+] T cells treated with very high (500 μM) levels of IPA displayed reduced cytokine production (Supplementary Fig. 5j, k). Furthermore, IPA treatment did not change the abundance of CD8[+] TCF1[+] stem-like cells in the metastatic liver of our model (Supplementary Fig. 5l, m), nor did it increase their abundance or TCF1 expression levels in vitro (Supplementary Fig. 5n–p).

## IPA reorganizes inflammatory immune cell interaction to indirectly boost T cell mediated anti-metastasis immunity

Since IPA did not directly impact T cell activation, we further characterized changes in the tumor immune microenvironment driven by IPA combining flow cytometry and highly multiplexed imaging mass cytometry (CyTOF) (Supplementary Fig. 6a). Flow cytometry analysis showed that the abundance and proportion of NKT cells, NK cells, macrophages, B lymphocytes, CD8[+] T cells,

dendritic cells and neutrophils were similar between the IPA and control treatment groups (Supplementary Fig. 6b, c). However, Ly6C[high] monocytes were reduced in IPA-treated liver, while the proportion of myeloid cells expressing MHCII was increased (Fig. 5a; Supplementary Fig. 6d). IPA treatment resulted in the accumulation of effector CD4[+] T cells with increased expression of IFNγ, TNFα and granzyme B (GZMB) (Fig. 5b–f; Supplementary Fig. 6e, f). We did not observe changes in the total liver CD8[+] T cell numbers (Supplementary Fig. 6b, c). However, IPA significantly increased the proportion of CD8[+] T cells expressing IFNγ, TNFα and GZMB (Supplementary Fig. 6g–i), indicating their enhanced antitumor activity.

Next, we assessed the spatial organization of immune cells in the metastatic liver microenvironment using highly multiplexed CyTOF imaging and identified the main resident and infiltrating immune populations in the liver along with hepatocytes, cancer cells, endothelial and biliary duct cells (Supplementary Fig. 6j, k, Supplementary Data 4)[40]. Using local spatial autocorrelation analysis, liver areas were divided into six distinct regions based on their cellular composition and organization (Fig. 5g; Supplementary Fig. 6l)[41]. Analysis of immune cell composition of metastasis regions revealed that IPA-treated mice displayed more total peri-metastatic CD4[+] T cells, while metastases of control mice contained more CD8[+] T cells, neutrophils (CD11b[+]Ly6g[+]) and monocytes (CD11b[+]Ly6g[-]F4/80[-]) (Fig. 5h). The immune cell composition of the non-tumor liver regions mirrored the composition of the tumor regions (Fig. 5h). Further characterization of CD4[+] T cell subsets revealed that IPA promoted an accumulation of inflammatory CD4[+]Tbet[+] (Th1) cells and reduced the abundance of immunosuppressive CD4[+]FOXP3[+] Tregs (Fig. 5i, j), aligning with flow cytometry results. Cell-cell interaction analysis in the metastatic regions showed that CD8[+] T cells were significantly and preferentially associated with CD4[+]Tbet[-] cells and Tregs in control tumors (Fig. 5k; Supplementary Fig. 6m), a spatial organization associated with reduced response to immunotherapy[42]. In contrast, while perimetastatic CD8[+] T cell numbers were decreased in IPA-treated liver metastasis, (Fig. 5h), they showed strong spatial colocalization with Th1 cells, an interaction that was absent in control livers (Fig. 5k; Supplementary Fig. 6m). This observation is functionally significant, as Th1 cells recruit cytotoxic CD8[+] T cells and promote their activation[43]. CD8[+] T cells did not interact with Tregs in IPA-treated metastases (Fig. 5k; Supplementary Fig. 6m).

Since IPA increased CD4[+] T cell activation and abundance in the perimetastatic region, we next asked if these cells were necessary for IPA-driven chemomemory. Therefore, we depleted CD4[+] T cells and assessed the ability of IPA to arrest liver metastatic growth (Fig. 5l). Analysis of blood and lymph nodes confirmed efficient CD4[+] T cell depletion (Supplementary Fig. 6n). CD4[+] T cell depletion abrogated the ability of IPA to suppress metastatic growth (Fig. 5m, n). Of note,

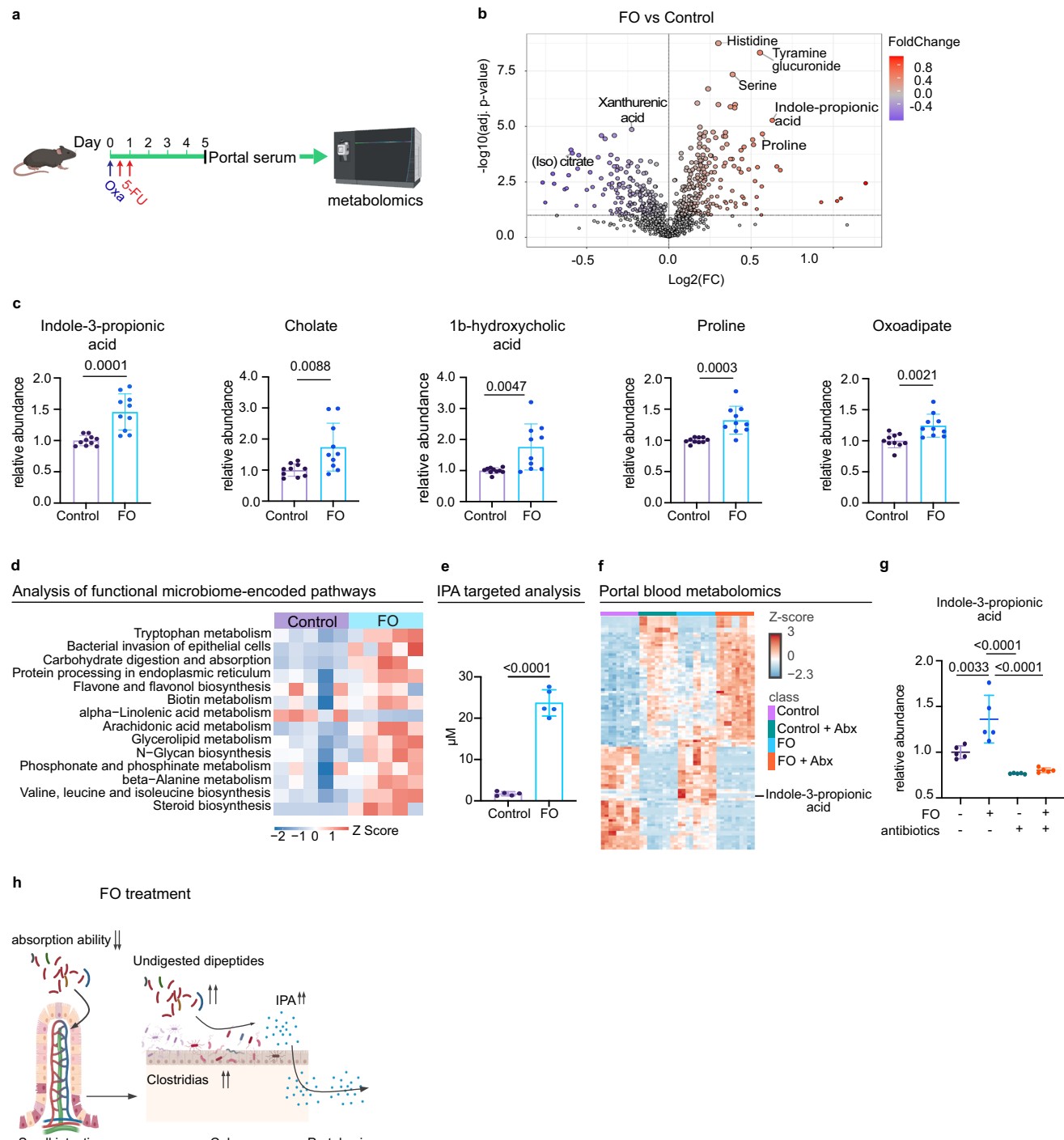

**Fig. 3 | 5-FU and oxaliplatin increase indole-3-propionic acid in portal vein circulation. a** Scheme of the experiment. Portal serum samples were analyzed by untargeted metabolomics 5 days post-PBS or FO treatment cycle. **b** Volcano plot of significantly upregulated (red; log2FC > 0.2, FDR < 0.05) and downregulated (blue; log2FC < −0.2, FDR < 0.05) metabolites in portal serum. Data representative of two independent experiments. **c** FO treatment increases tryptophan-derived metabolite, indole-3-propionic acid and other metabolites. Data are pooled from two independent experiments. $n = 10$ per condition. **d** FO treatment increases bacterial tryptophan metabolism. Functional analysis of bacterial communities using Picrust2; heatmap of significantly changed KEGG pathways. **e** Concentration of portal serum IPA in PBS and FO-treated mice 5 days post treatment, measured by

RPLC-MS/MS. $n = 5$ per condition. **f** Heatmap showing metabolite abundance changed in the portal serum of microbiota-intact and microbiota-depleted mice treated with FO or PBS. **g** IPA is one of the most reduced metabolites in microbiota-depleted mice. Relative abundance of indole-3-propionic acid in the indicated conditions, $n = 5$ per condition. **h** Model of FO-induced dysbiosis. Data are shown as mean ± SD and analyzed using two-tailed unpaired Student's t-test (**e**) with multiple testing correction with the BH method (**c**), from metabolite consistently increased (adjusted-p-value < 0.05) in two independent experiments. Data are shown as mean ± SD and analyzed using one-way Anova with Tukey's multiple comparisons test (**g**). Icons were created in *BioRender under the license SABINE, A. (2025)* https://BioRender.com/ pmy8rs7. Source data are provided as a Source Data file.

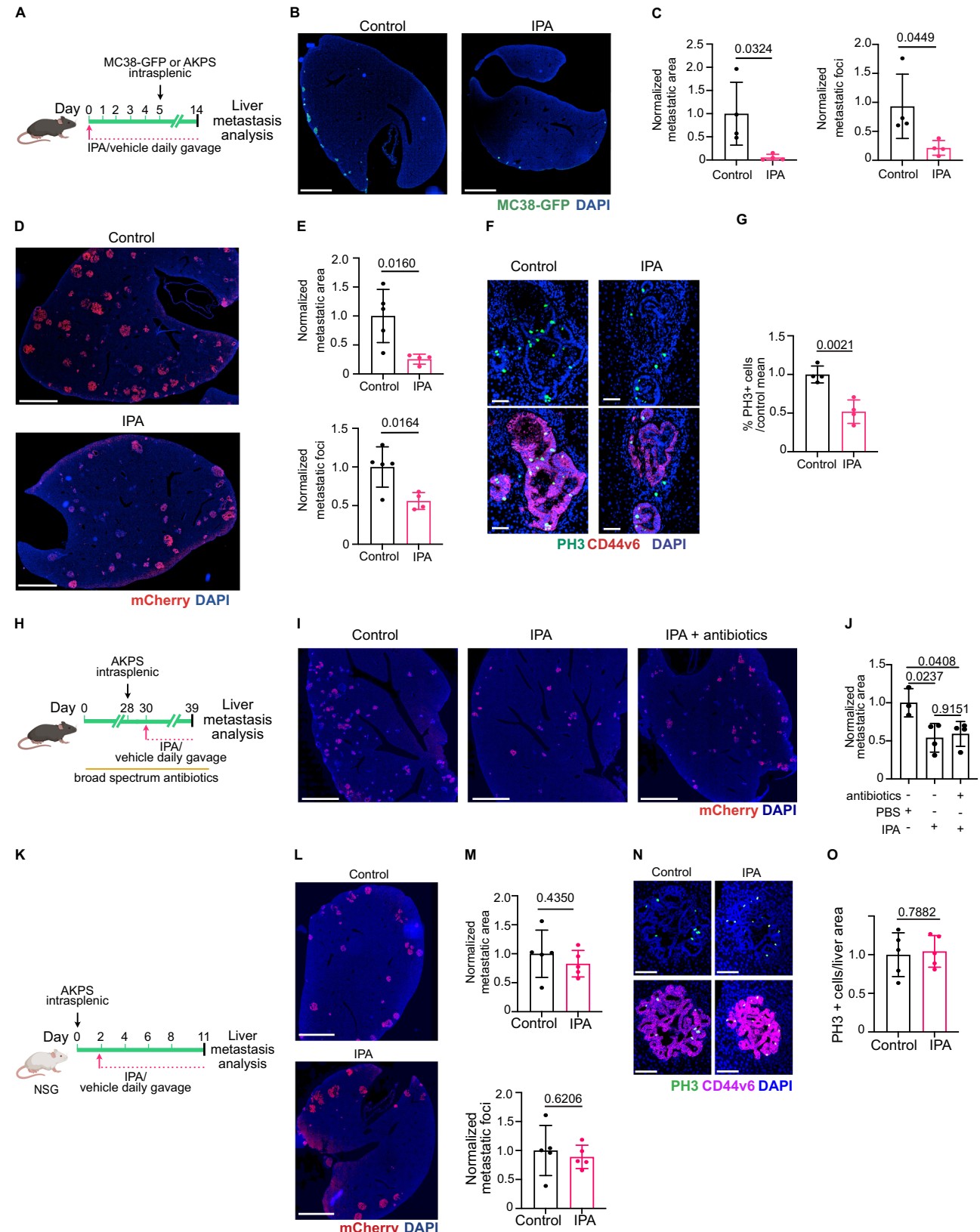

the absence of CD4[+] T cells in control non-IPA treated mice reduced the metastatic burden compared to IPA-treated mice (Fig. 5m, n), likely due to the depletion of regulatory CD4[+] T cells in control mice. The infiltration of CD8[+] T cells was similar between treatment groups (Supplementary Fig. 6o, p). These results show that IPA drives an indirect CD4[+] T cell antitumor effect.

## IPA reduces Ly6C[high] monocyte abundance promoting CD4[+] T cell-driven antitumor immunity

Myeloid cells play a key role in CD4[+] T cell priming and we observed an increased proportion of myeloid cells expressing MHCII in the livers of IPA-treated mice (Supplementary Fig. 6d), while perimetastatic monocytes and total liver Ly6C[high] monocytes were reduced

**Fig. 4 | Exogenous IPA supplementation recapitulates 5-FU and oxaliplatin anti-metastatic effect through immune-mediated mechanism. a** Scheme of the experiment. Mice were gavaged daily with IPA or PBS 5 days before MC38-GFP or AKPS intrasplenic injection and until the end of the experiment. **b** IPA inhibits liver metastasis in MC38 model. Staining for cancer cells (GFP, green); DNA (blue). Scale bar: 2 mm. **c** Quantification of metastasis area and number. Data normalized to liver area and control mean. $n = 4$ per condition. **d** IPA inhibits liver metastasis in AKPS model. Staining for cancer cells (mCherry, red) and DNA (blue). Scale bar: 2 mm. **e** Quantification of metastasis area and number. Data normalized to liver area and control mean. Control, $n = 5$. IPA, $n = 4$. **f** IPA inhibits cancer cell proliferation. Staining for phospho-histone3 (PH3, green), colon cancer stem cells (CD44v6, magenta) and DNA (blue). Scale bar: 50 μm. **g** Quantification of cancer cell proliferation. Data normalized to control mean. $n = 4$ per condition. **h** Scheme of the experiment to analyze the effect of IPA in microbiota-depleted mice. **i** IPA inhibits

metastasis in the absence of microbiota. Staining for cancer cells (mCherry, red) and DNA (blue). Scale bar: 2 mm. **j** Quantification of metastatic area per liver area normalized to control mean. Control, $n = 3$. IPA, $n = 4$. Antibiotics + IPA, $n = 4$. **k** Scheme of the experiment. NSG mice were injected with AKPS organoids and received daily IPA or PBS gavage. **l** IPA is ineffective in immunocompromised mice. Staining for mCherry (red) and DNA (blue). Scale bar: 2 mm. **m** Quantification of metastatic area and number per liver area normalized to control mean. $n = 5$ per condition. **n** Staining for phospho-histone3 (PH3, green), CD44v6 (magenta) and DNA (blue). Scale bar: 100 μm. **o** Quantification PH3+ cells per metastatic area normalized to control mean. $n = 5$ per condition. Data are shown as mean ± SD and analyzed using two-tailed unpaired Student's t-test (**c, e, g, m, o**) and one-way Anova with Tukey's multiple comparisons test (**j**). Icons were created in *BioRender under the license SABINE, A. (2025)* https://BioRender.com/ pmy8rs7. Source data are provided as a Source Data file.

(Fig. 5a, h)[44,45]. Ly6C[high] monocytes are immunosuppressive and reduce T cell activation while also promoting Treg differentiation[46,47]. Metastatic liver Ly6C[high] monocytes abundance negatively correlated with the presence of CD4+ T cells (Fig. 5o) but not with other major immune cell subtypes including CD8+ T or CD19+ B cells (Supplementary Fig. 6q, r). We therefore hypothesized that IPA-mediated loss of Ly6C[high] monocytes was driving CD4+ T cell antitumor effect in liver metastases. Consequently, we directly assessed the role of Ly6C[high] monocytes in the suppression of CD4+ T cell activation in vitro. We isolated Ly6C[high] monocytes from the metastatic liver and bone marrow of metastasis-bearing and control mice, as normal liver does not contain sufficient amount of these cells. We then co-cultured these cells with CD4+ T cells at a ratio of 1:4 to replicate the conditions present in the metastatic liver (Fig. 5p). Tumor-associated, but not bone marrow derived, Ly6C[high] monocytes significantly inhibited CD4+ T cell proliferation in vitro (Fig. 5q, r). These data showed that an IPA-driven reduction in Ly6C[high] monocytes abundance is sufficient to promote an indirect CD4+ T cell antitumor effect in the metastatic liver.

In all, these in vivo analyses indicate that IPA reorganizes the immune microenvironment of the metastatic liver by reducing the infiltration of immunosuppressive Ly6C[high] monocytes while increasing proinflammatory MHCII-expressing myeloid cell entry. This in turn promotes the expansion of proinflammatory Th1 cells in the metastatic niche, ultimately leading to their enhanced interaction with cytotoxic CD8+ T cells in the peritumoral region and promotion of a CD4+ T cells mediated antitumor effect (Fig. 5s).

## IPA skews bone marrow myeloid progenitor differentiation to produce macrophages in lieu of Ly6C[high] monocytes

Given that the reduction of Ly6C[high] monocytes resulted in a CD4+ T cell-mediated antitumor effect, we investigated how IPA reduces the Ly6C[high] population. Monocytes are short-lived in the tumor microenvironment, and their production and maintenance are tightly regulated by bone marrow hematopoiesis[48]. Flow cytometry analysis of blood revealed a significant reduction of circulating Ly6C[high] monocytes in IPA-treated mice (Fig. 6a). However, consistent with observations in the metastatic liver (Supplementary Fig. 6b, c), IPA did not change circulating numbers of neutrophils, macrophages (CD11b+F4/80+), CD4+ and CD8+ T cells (Supplementary Fig. 7a, b).

The decreased abundance of Ly6C[high] monocytes both in the blood and the liver suggests that IPA rewires the production of myeloid cells in the bone marrow. IPA did not alter the proportion of hematopoietic stem (HSC) cells, multipotent progenitor (MPP) cells, common lymphoid progenitor (CLP) cells, common myeloid progenitor (CMP) cells, megakaryocyte/erythroid progenitor (MEP) cells and granulocyte-monocyte progenitor (GMP) cells (Supplementary Fig. 7c). We tested if IPA treatment affected bone marrow capacity to produce myeloid cells. We isolated bone marrow cells from liver metastasis-bearing mice treated with control or IPA and tested the

colony-forming ability of myeloid progenitors in a methylcellulose assay (Fig. 6b). We found that bone marrow cells from IPA-treated mice displayed significantly reduced myeloid colony forming units (CFU) compared to controls (Fig. 6c). To determine if IPA reduces the production of Ly6C[high] monocytes, we treated bone marrow cells with immunosuppressive myeloid cell-inducing media in the presence of vehicle or IPA (10 μM or 50 μM) for 5 days and analyzed resulting populations by FACS. At 10 μM, IPA did not affect overall cell viability or cellularity, whereas at 50 μM viability was slightly reduced (Fig. 6d). Most importantly, IPA significantly reduced the abundance of Ly6C[high] monocytes in a dose-dependent manner (Fig. 6e), while the abundance of neutrophils was unchanged by IPA treatment (Supplementary Fig. 7d, e). Thus, these results show that IPA impacts myeloid progenitor ability to produce myeloid cell and Ly6C[high] monocytes in vitro and in vivo.

To decipher how IPA blunts Ly6C[high] monocytes production, we performed RNAseq on bone marrow cells cultured in immunosuppressive myeloid cell-inducing media in the presence of vehicle or physiological and supraphysiological levels (10 μM and 50 μM, respectively) of IPA[37]. As 50 μM of IPA reduced cell viability, we focused our analysis on the 10 μM treatment. Differential expression analysis revealed 105 upregulated and 18 downregulated genes in IPA-treated bone marrow cells (Fig. 6f; Supplementary Data 5). Unbiased gene set enrichment analysis (GSEA) showed treatment with IPA led to concerted suppression of transcripts characteristic of immunosuppressive myeloid cell[49], consistent with flow cytometry results (Fig. 6g). Furthermore, IPA increased pathways related to chemotaxis, macrophage activation, phagocytosis, antigen presentation and leukocyte differentiation (Fig. 6h). Notably, transcripts associated with macrophage differentiation (*Cd68, Mafb, Flrt2, Atp6v0d2*) and polarization (*Arg1, Spp1, Ecm1, Epsti1, Dusp3*) were upregulated by IPA (Fig. 6f)[50–52]. In contrast, expression of genes associated with immature myeloid cells, myeloid progenitor cells and differentiation regulators were reduced (*Ms4a3, Chil3, Lcn2, Pak1, Klf5*) (Fig. 6f)[53–55]. Although indole-derived metabolites are often linked to AHR signaling, the AHR transcriptional signature remained unchanged following IPA treatment, which is consistent with inability of IPA to activate AHR in biochemical assays (Supplementary Fig. 7f and Supplementary Data 5)[38]. Taken together, these data indicated that IPA accelerates myeloid cell maturation towards macrophages thus reducing the abundance and release of Ly6C[high] monocytes. In line with this hypothesis, pathways suppressed by IPA were linked with epigenetic changes such as methylation, histone modification but also reduction in DNA repair, aligning with a transition towards a more differentiated cell state (Fig. 6h)[56,57].

To determine underlying gene regulatory networks responding to IPA treatment, we used the SCENIC algorithm adapted to bulk RNAseq analysis[58,59]. *Rbpj, Mafb, Irf8, Klf4, Mitf* and *Elk3* regulons were significantly enriched in IPA-treated samples, while *Elf4, Maz* and *Fos* regulons were more active in control cells (Supplementary Fig. 7g). Transcriptional activity of

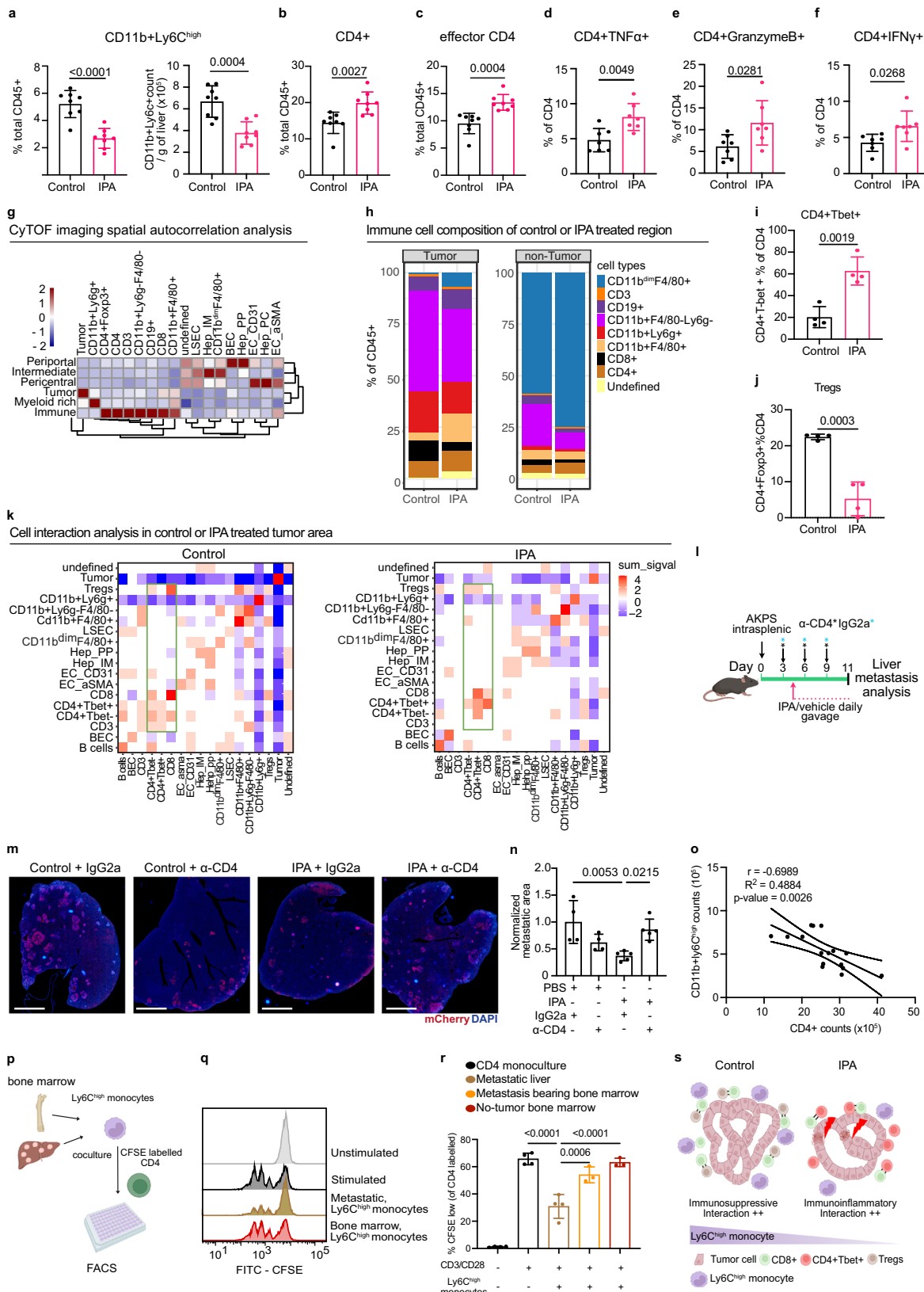

**g** CyTOF imaging spatial autocorrelation analysis

**h** Immune cell composition of control or IPA treated region

**k** Cell interaction analysis in control or IPA treated tumor area

*Rbpj*, *Mafb*, *Irf8* and *Klf4* have been shown to promote myeloid cell maturation and macrophage differentiation[50,51,60–64], while *Mitf* is essential for osteoclast differentiation downstream of M-CSF[65]. Conversely, *Fos* mutant mice accumulate macrophages in their bone marrow and *Fos* activity is reduced by IPA[66]. Furthermore, *Elf4* and *Maz* regulons are downregulated by IPA and

are important in hematopoietic progenitor maintenance and cell differentiation[67–69].

## IPA targets common myeloid progenitors fate

The bone marrow cell analysis of control and IPA-treated mice bearing metastases revealed that IPA treatment significantly increased

**Fig. 5 | IPA indirectly boosts T cell activation driving anti-metastasis immunity by reducing the abundance of immunosuppressive Ly6C$^{high}$ monocytes. a** IPA reduces Ly6C$^{high}$ monocyte infiltration in the metastatic liver. **b** IPA increases CD4$^+$ T cells. CD4$^+$ cell percentage of total CD45$^+$ cells. **c** IPA increases CD4$^+$ CD44$^+$ Teff cells. **a–c** $n$ = 8 per condition, from two independent experiments. IPA promotes accumulation of Th1-like and cytolytic CD4$^+$ T cells expressing (**d**) TNFα, (**e**) granzyme B and (**f**) IFNγ. $n$ = 7 per condition. **g** Cell compositions of each region identified by local spatial autocorrelation from CyTOF imaging. **h** Immune composition of peritumoral (tumor cells, myeloid, and inflammatory) and non-tumoral (tumor-free periportal, intermediate and pericentral) liver regions. **i** IPA increases liver infiltration with Th1-like CD4$^+$Tbet$^+$ cells. $n$ = 4 per condition. **j** IPA decreases liver infiltration with CD4$^+$ FOXP3$^+$ Tregs. $n$ = 4 per condition. **k** Heatmaps show cellular colocalization in control versus IPA-treated samples. **l** Scheme of the experiment to test effect of CD4$^+$ T cell depletion on IPA-mediated metastasis inhibition.

**m** Staining for cancer cells (mCherry, red) and DNA (blue). Scale bar: 2 mm. **n** Quantification of metastatic area per liver area normalized to control mean. Control + IgG2a, $n$ = 4. Control + αCD4, $n$ = 4. IPA + IgG2a, $n$ = 5. IPA + αCD4, $n$ = 5. **o** Ly6C$^{high}$ monocyte abundance anti-correlates with the presence of CD4$^+$ T lymphocytes. $n$ = 8 per condition. **p** Scheme of the Ly6C$^{high}$ monocytes and CD4$^+$ T cell coculture experiment. **q** Histogram of CFSE dilution of labeled CD4$^+$ T cells in the indicated conditions. **r** Quantification of CFSE low proliferated CD4$^+$ T cells. Data represent 4-3 biological replicates per condition, representative of two 2 independent experiments. **s** The effect of IPA on the metastatic immune microenvironment. Data are shown as mean ± SD and analyzed using two-tailed unpaired Student's t-test (**a–f, i, j**), Pearson correlation test (**o**) and one- way Anova with Tukey's multiple comparisons test (**n, r**). Icons were created in *BioRender under the license SABINE, A. (2025)* https://BioRender.com/ pmy8rs7. Source data are provided as a Source Data file.

frequency of macrophages while the total abundance of neutrophils and overall monocytes remained unchanged (Fig. 6i), supporting the increase macrophages maturation in the bone marrow.

To study the underlying mechanisms, we further analyzed cell cycle of CMPs and GMPs from control or IPA-treated mice. IPA significantly increased the proportion of quiescent CMPs compared to GMPs (Supplementary Fig. 7h), suggesting CMPs are the target of IPA.

To further investigate how IPA affects CMP differentiation, we performed a CMP adoptive transfer experiment. We induced metastases in CD45.1 mice and treated them with either control or IPA daily. We then FACS-sorted CMPs from bone marrow and transferred them into CD45.2 recipient mice (Fig. 6j), tracking CMP fate at early time points in vivo using established protocols[70,71]. The proportion of CD45.1-derived bone marrow cells and the frequency of monocytes from transferred CMPs were similar between groups (Fig. 6k). In contrast, CMPs from IPA-treated mice generated a higher proportion of macrophages in the bone marrow and fewer CCR2-expressing monocyte (Fig. 6k). Consistent with the key role of CCR2 in monocyte egress from bone marrow, we observed fewer CD45.1-derived monocytes in the blood of mice transferred with CMPs from IPA-treated donors (Fig. 6l)[72].

Collectively, we propose that post-chemotherapy microbiota remodeling and increased IPA production play a central role in establishing a metastasis-refractory state. IPA primarily targets CMPs in the bone marrow, promoting their differentiation toward the macrophage lineage while limiting the release of Ly6C$^{high}$CCR2$^+$ monocytes into circulation. This normalization of tumor-induced pathological myelopoiesis reduces circulating monocytes available for tumor infiltration, thereby enhancing anti-tumor T cell responses (Fig. 6m).

### IPA potentiates chemotherapy response against CRC liver metastasis

To test the translational relevance of our observations we sought next to determine if IPA supplementation improves response to chemotherapy in mice harboring growing liver metastasis (Fig. 7a). We injected AKPS organoids intrasplenically and administered PBS or IPA orally from day three until the end of the experiments and treated mice with two cycles of FO with a one-week interval (Fig. 7a). Cancer cell proliferation was reduced in IPA-treated metastases as compared to controls as determined by staining for phosphorylated histone H3 (PH3) (Fig. 7b, c). Furthermore, combinatorial FO + IPA therapy significantly decreased metastatic burden and increased necrotic areas within liver metastatic lesions (Fig. 7d–g). Assessment of immune cell infiltration showed that IPA treatment increased CD4$^+$ T cell infiltration into metastatic lesions both when used as monotherapy and in combination with FO, whereas FO alone had no significant effect (Fig. 7h, i). CD8$^+$ T cell infiltration was not significantly

modified in any treatment (Fig. 7j), further confirming that IPA mostly indirectly affects CD4$^+$ T cells (Fig. 7i, j).

### Chemotherapy increases IPA concentration and IPA negatively correlates with peripheral blood monocytes in CRC patients

To test the relevance of our observations from animal models to human disease, we recruited fourteen CRC patients with primary only or metastatic disease that were scheduled to receive adjuvant chemotherapy (Fig. 8a; Supplementary Table 1). Patients were chemotherapy naïve to reduce the interference of previous treatment exposure. Serum samples were collected before the initiation of the first cycle of chemotherapy (T1) and just before the initiation of the second cycle of the chemotherapy regimen (T2) (Fig. 8a). Of note, only two CRC patients were not treated with antibiotics within the last two months before blood collection (Supplementary Table 1). For control baseline values, we sampled serum from sex and partially age-matched healthy controls from the SwissChronoFood study, who did not receive antibiotics treatment within 2 months preceding blood collection (Supplementary Table 1)[73]. Targeted metabolomic analyses was carried out to quantify concentrations of tryptophan, a tryptophan-derived bacterial metabolite and precursor of IPA, 3-indolelactic acid (ILA) and IPA (Fig. 8b)[33]. When comparing the healthy cohorts to baseline values of CRC patients (T1), tryptophan concentration was similar between the two cohorts (Fig. 8c). A slight 20%, yet statistically significant, decrease in the concentration of ILA, was observed in the CRC cohort (Fig. 8c). Strikingly, IPA concentration in baseline (T1) CRC patient sera were nearly fivefold lower compared to control sera (Fig. 8c). IPA was undetectable in two patients and IPA levels were not correlated with age (Supplementary Fig. 8a). Although further studies are necessary, lower IPA levels in CRC patients may reflect depletion of IPA-producing bacteria by commonly used antibiotics.

When assessing the effect of chemotherapy, we found that circulating tryptophan and ILA levels were similar pre- and post-chemotherapy (T1 vs T2) (Fig. 8d). In contrast, IPA was significantly increased at T2 in 9 out of 14 patients as compared to IPA baseline concentration (T1), while it remained undetectable in two CRC patients (Fig. 8d). IPA concentration reached the healthy population concentration in 4 patients (Fig. 8d). Circulating IPA, but not tryptophan and ILA concentration, significantly and negatively correlated with monocyte abundance in the peripheral blood of CRC patients (Fig. 8e; Supplementary Fig. 8b–e). In contrast, IPA levels did not correlate with circulating neutrophils and lymphocytes (Fig. 8f, g). Most patients with substantial increase in IPA level at T2 showed a corresponding reduction of monocyte at T2 (Supplementary Fig. 8f, g) and IPA concentration positively correlated with the lymphocyte-to-monocyte ratio (Fig. 8h), a parameter associated with improved overall survival in CRC patients[74]. These results are consistent with our conclusions in animal models that IPA influences myeloid progenitors and limits tumor by reducing circulating levels of immunosuppressive monocytes. Importantly, survival analysis of 1024 CRC patients with localized CRC from

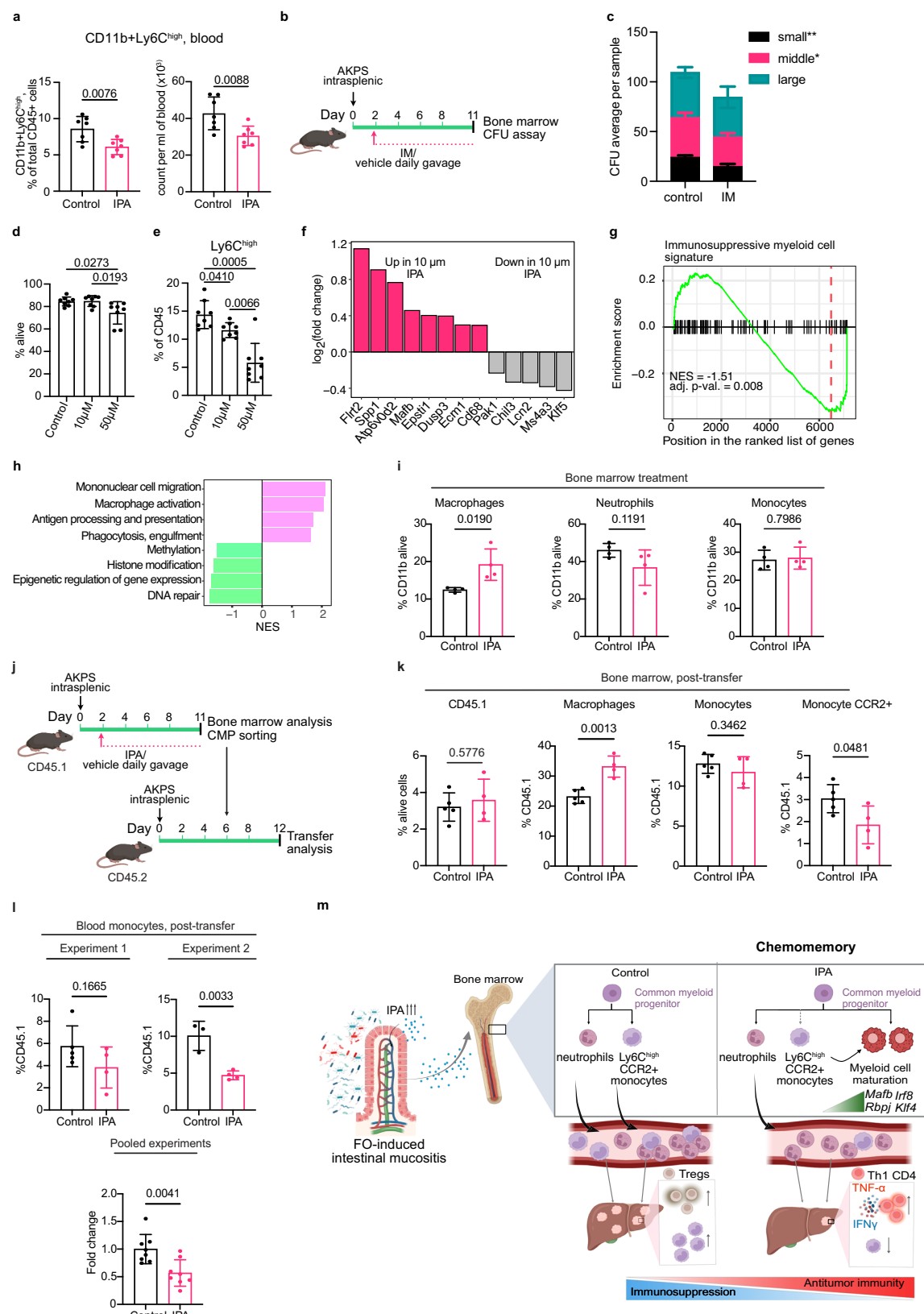

the institutional database of the University Hospital of Geneva (HUG) showed that high monocyte concentration is associated with a worse overall survival (Fig. 8i), demonstrating the importance of reducing monocyte counts and the potential of using IPA to improve overall survival in patients with CRC. In all these data demonstrate that IPA-mediated chemomemory effect is observed in patients and IPA concentration is directly anti-correlated with monocyte abundance in CRC patients.

## Discussion

Direct anti-tumor action of chemotherapy requires the microbiota, and the presence of specific taxa predict response to treatment[5,6,75].

**Fig. 6 | IPA limits myeloid progenitor ability to produce immunosuppressive Ly6C$^{high}$ monocytes. a** IPA reduces circulating Ly6C$^{high}$ monocytes. $n = 7$ per condition. **b** Experiment scheme. **c** CFU counts from bone marrow of control and IPA-treated mice. $n = 4$ per condition in duplicate. **d** Physiological IPA concentration does not affect bone marrow cell viability. Quantification of CD45$^+$ 7AAD- cells. $n = 8$ per condition. Biological replicates pooled from two independent experiments. **e** IPA inhibits generation of Ly6c$^{high}$ monocytes in vitro. Percentage of Ly6C$^{high}$ monocytes generated with vehicle or IPA in vitro. $n = 8$ per condition. Biological replicates pooled from two independent experiments. **f** IPA induces macrophage-related genes. Barplots of selected transcripts significantly upregulated (pink) or downregulated (gray) during in vitro myeloid differentiation with vehicle or 10 μM IPA. **g** IPA reduces immunosuppressive myeloid cell gene signature. **h** IPA upregulates pathways linked to macrophage maturation and function. **i** IPA increases the macrophages abundance in the bone marrow of metastasis-bearing mice. $n = 4$ per condition. **j** Scheme of the experiment to analyze the fate of CMPs treated with

IPA in vivo. **k** CMPs from IPA-treated mice produce more macrophages in the bone marrow of recipient mice. Data are representative of two independent experiments. Control, $n = 5$. IPA, $n = 4$. **l** CMPs from IPA-treated mice release less monocytes in the blood. Experiment 1: Control, $n = 5$. IPA, $n = 4$; Experiment 2: Control, $n = 3$. IPA, $n = 4$. **m** IPA normalizes pathological myelopoiesis. In the absence of IPA, tumor-induced Ly6C$^{high}$CCR2$^+$ monocytes are released from the bone marrow and recruited to the tumor microenvironment, where they suppress antitumor immunity. IPA promotes monocyte maturation, limiting their release into circulation. This relieves antigen-presenting cell immunosuppression and enhances CD4$^+$ Th1-mediated antitumor immunity in the liver metastatic niche. Data are shown as mean ± SD and analyzed using two-tailed unpaired Student's t-test (**a, c, i, k, l**), repeated measure one-way Anova with Tukey's multiple comparisons test (**d, e**). Icons were created in *BioRender under the license SABINE, A. (2025)* https://BioRender.com/ pmy8rs7. Source data are provided as a Source Data file.

Groundbreaking work demonstrated that myeloid cell intratumoral inflammatory responses, which promote the acute antineoplastic effect of oxaliplatin, are microbiota dependent[5]. Vaccinating mice with oxaliplatin-induced apoptotic ileal epithelial cells provides lasting protection against subcutaneous CRC tumors in mice[12]. Here, we report that prior history of 5-FU and oxaliplatin treatment durably limits liver CRC metastasis formation in mice. This chemomemory was independent of a direct effect of chemotherapy on tumor cells. Instead, we found that chemomemory was driven by chemotherapy-induced intestinal mucositis, modifying nutrient availability to gut bacteria which in turn drove intestinal dysbiosis. Chemotherapy-induced dysbiosis was associated with the expansion of tryptophan-metabolizing bacteria and production of indole-3-propionic acid (IPA). We further demonstrated that IPA oral supplementation is sufficient to limit CRC liver metastasis formation both in preventive and therapeutic settings. Mechanistically, IPA shifted bone marrow myeloid progenitor differentiation from a program producing immature myeloid cells, precursors for Ly6C$^{high}$ monocytes, towards a mature macrophage differentiation program. This shift in bone marrow progenitor differentiation reduced the number of circulating and hepatic Ly6C$^{high}$ monocytes, which increased antitumor CD4$^+$ T cells and, in turn, inhibited the formation of liver metastases.

A typical chemotherapy cycle lasts two weeks, and patients usually undergo 6 to 12 cycles of FOLFOX treatment. We found that the metastasis-refractory state lasts up to 10–20 days after a single FO cycle (Fig. 1l–n; Supplementary Fig. 2b–e). It will be important to determine in future studies whether repeated cycles lead to sustained high levels of IPA, prolonged immune system rewiring, and potentially stronger control of metastasis.

Our work emphasizes the importance of considering cancer as a systemic disease and highlights the importance of understanding normal organ responses for predicting and enhancing the effectiveness of anti-cancer therapies. The findings carry several significant implications: 1) chemotherapy-induced dysbiosis and bacterial-metabolite-mediated signaling within the bone marrow should be taken into account as pivotal regulators of the pre-metastatic niche; 2) broad-spectrum antibiotics—commonly administered during primary CRC tumor resection—or an unfavorable microbiota, before or after chemotherapy may promote metastasis development; 3) exogenous IPA supplementation, in combination with standard-of-care regimens, has the potential to enhance therapeutic efficacy in patients with metastatic CRC. The high variability of IPA levels observed in patients suggests the presence of other patient-specific factors that may limit this protective effect. These include differences in microbiota composition and function, diet, and concomitant medications, all of which can result in insufficient IPA production. Understanding these variables could inform the development of more personalized adjuvant strategies based on microbial products.

While this work was in progress, Jia et al. reported that IPA potentiated immunotherapy in multiple cancer models by inducing H3K27 acetylation at the super-enhancer region of *Tcf7*, reprogramming CD8 T cell stemness[8]. In our setting, IPA exerted a more pronounced and direct effect on myeloid cells rather than T lymphocytes, aligning with another study showing that physiological IPA concentrations do not alter CD4$^+$ T cell activity[76]. Notably, despite a marked antitumor effect, IPA did not expand TCF1$^+$ stem-like CD8 T cells in the metastatic liver microenvironment (Supplementary Fig. 5l, m) nor did it modify their production in vitro (Supplementary Fig. 5l–p). In contrast, physiological IPA concentrations significantly reduced the production of immunosuppressive Ly6C$^{high}$ monocytes in vitro and in vivo (Figs. 5a and 6e). Numerous previous studies have documented that reducing Ly6C$^{high}$ monocytes abundance enhances antitumor immunity across various solid tumors models[46,77]. The spatial distribution and cell-cell contacts of immune cells within the tumor microenvironment are of paramount importance for productive antitumor immunity[45,78,79]. We found that this reduction enhanced the infiltration and antitumor activity of CD4$^+$ T cells by promoting Th1 differentiation and reorganizing the spatial positioning and interactions CD8$^+$ T cells with CD4$^+$ T cells within the metastatic liver microenvironment.

Mechanistically, IPA drove myeloid cell differentiation toward a macrophage fate, thereby reducing the abundance of Ly6C$^{high}$ monocytes. Analysis of underlying transcriptional networks revealed a concerted shift in several pathways linked to macrophage differentiation. Notably, IPA activated Rbpj-driven regulons suggesting a role of Notch pathway in IPA-mediated myeloid cell maturation. Notch signaling is crucial for the maturation of myeloid cells, and recent work has shown that Rbpj is necessary to convert monocytes to macrophages in liver disease[80–82]. Although IPA has been suggested to act as a ligand of AHR[36,83], AHR signature was not modified by IPA (Supplementary Fig. 7f and Supplementary Data 5). Furthermore, AHR activation has been linked to pro-tumoral functions of myeloid cells, whereas we observed the opposite effect[84]. Thus, while additional studies are needed to identify the molecular effectors of IPA in myeloid progenitor cells, our results reveal a role of IPA in myeloid cell differentiation.

Our work has several translational implications. First, loss of IPA upon antibiotic treatment in mice and low levels of IPA in CRC patients, most of whom were treated with broad spectrum antibiotics, reinforce the notion that microbiota and in particular tryptophan-metabolizing bacteria are essential for the effectiveness of anti-cancer drugs[85,86]. Second, IPA levels were increased by chemotherapy in a subset of CRC patients and IPA levels negatively correlated with circulating levels of monocytes but not neutrophils and lymphocytes. Interestingly, microbiota depletion reduced cisplatin efficacy in an ovarian cancer model while microbiota competent mice responding to cisplatin displayed increased IPA levels[87]. Larger cohort of patients and prospective follow up study are now required to determine if IPA levels post- chemotherapy alone or in

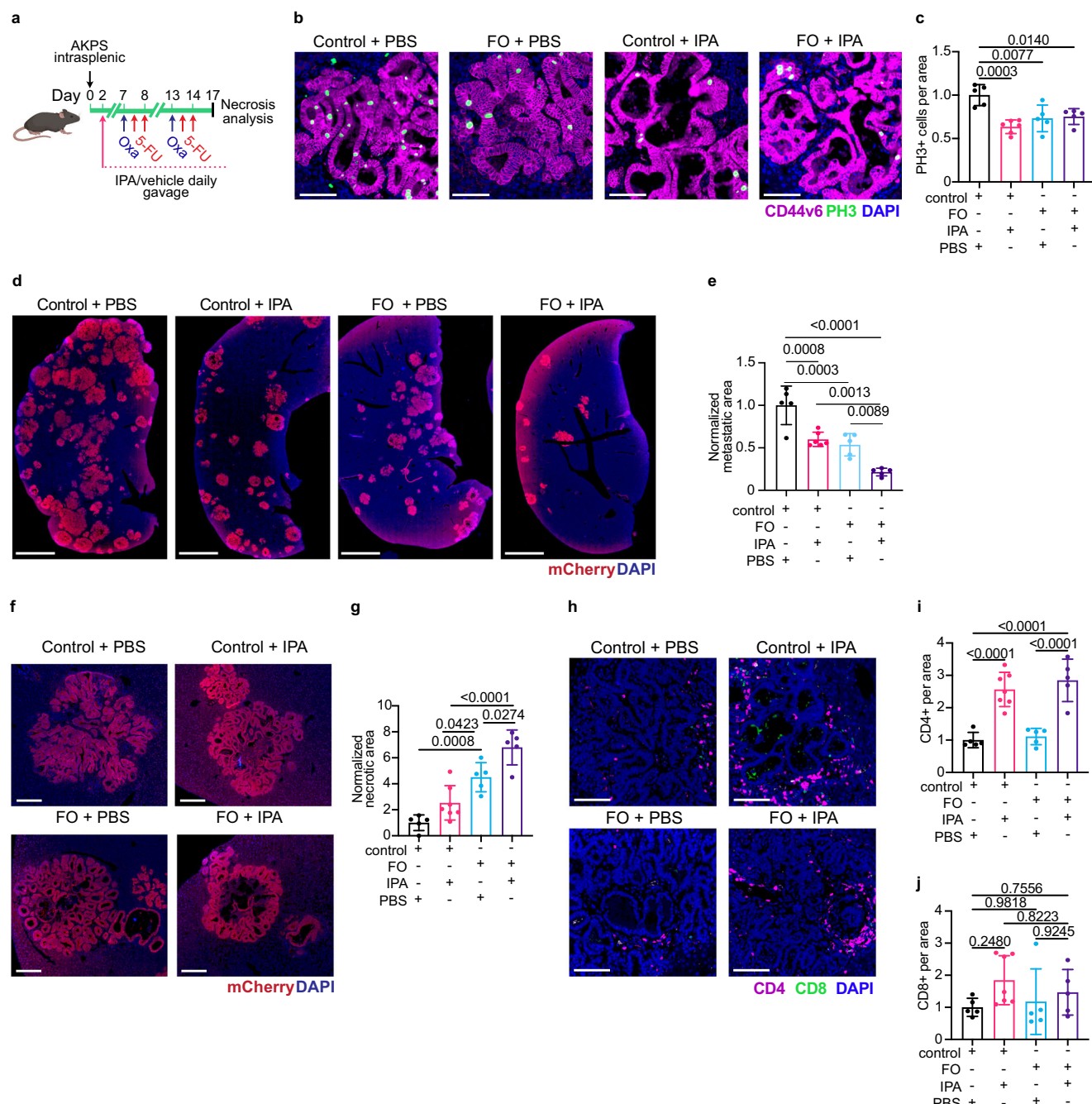

**Fig. 7 | IPA potentiates chemotherapy response of CRC liver metastasis.**
**a** Scheme of the experiment to analyze the impact of IPA supplementation on chemotherapy responses. **b** Both IPA supplementation and FO decrease cancer cell proliferation. Staining for phospho-histone 3 (PH3, green), cancer cells (CD44v6, red) and DNA (DAPI, blue). Scale bar: 100 μm. **c** Quantification of PH3⁺ cells within metastatic lesion normalized to control mean. Control + PBS, $n = 5$. Control + IPA, $n = 6$. FO + PBS, $n = 5$. FO + IPA, $n = 5$. **d** IPA supplementation potentiates FO and reduces liver metastasis load. Staining for cancer cells (mCherry, red) and DNA (DAPI, blue). Scale bar: 2 mm. **e** Quantification of metastasis area. Data normalized to liver area and control mean. Control + PBS, $n = 5$. Control + IPA, $n = 6$. FO + PBS, $n = 5$. FO + IPA, $n = 5$. **f** IPA supplementation potentiates FO and increases tumor necrosis.

Staining for cancer cells (mCherry, red) and DNA (blue). Scale bar: 200 μm. **g** Quantification of necrotic area within metastatic lesion normalized to control mean. Control + PBS, $n = 5$. Control + IPA, $n = 7$. FO + PBS, $n = 5$. FO + IPA, $n = 5$. **h** IPA supplementation potentiates tumor CD4⁺ T cell infiltration. Staining for CD4 (magenta), CD8 (green) and DNA (blue). Scale bar: 100μm. **i** Quantification of CD4⁺ T cells and **j** CD8⁺ T cells per metastatic area normalized to control mean. Control + PBS, $n = 5$. Control + IPA, $n = 7$. FO + PBS, $n = 5$. FO + IPA, $n = 5$. Data are shown as mean ± SD and analyzed using one-way Anova with Tukey's multiple comparisons test (**c, e, g, i, j**). Icons were created in *BioRender under the license SABINE, A. (2025)* https://BioRender.com/ pmy8rs7. Source data are provided as a Source Data file.

combination with analyses of circulating Ly6C^high monocytes are prognostic for progression-free and overall survival in CRC patients and other cancer types.

Our results have implications beyond cancer, such as in the context of aging-associated increases in myelopoiesis or in chronic

inflammatory diseases. For example, young mouse microbiota transplantation blunted myelopoiesis in aged mice and this was associated with elevated circulating levels of the tryptophan-derived microbial metabolites indole-3-carbinol and IPA[88]. Furthermore, IPA supplementation reduced the abundance of lung monocytes in an asthma

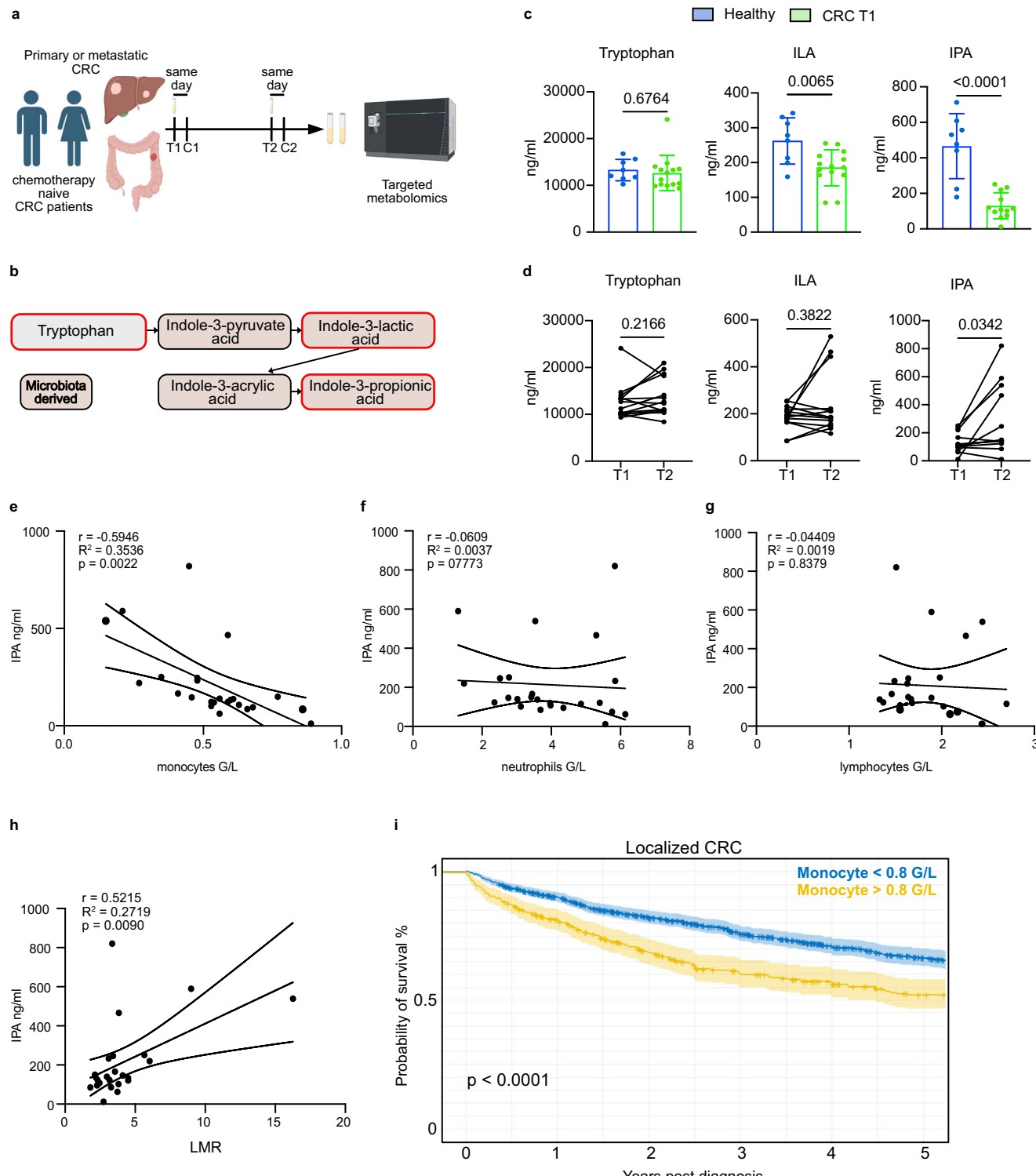

**Fig. 8 | Chemotherapy increases IPA concentration in subset of chemo-naïve CRC patients. a** Patient recruitment scheme. Blood was sampled in chemotherapy-naïve CRC patients before the first cycle (T1) and before the initiation of their second cycle (T2) of chemotherapy. C1, C2: chemotherapy cycles 1 and 2. **b** Pathway of tryptophan conversion to indole-3-propionic acid. Microbial-derived metabolites are shown in brown. Red lines: metabolites analyzed by targeted metabolomics. **c** Circulating IPA levels are reduced in CRC patients at baseline as compared to healthy population. Targeted metabolomic analysis of tryptophan, indole-3-lactic acid (ILA) and IPA in the indicated groups. Healthy, *n* = 8. CRC T1, *n* = 14 for tryptophan and ILA, *n* = 12 for IPA. **d** IPA is significantly increased post-chemotherapy in CRC patients. Tryptophan, ILA and IPA concentrations at T1 and

T2 in CRC patients. *n* = 12-14. IPA was undetectable in two patients. **e** IPA blood concentration anti-correlates with blood monocyte counts in CRC patients at T1 and T2. **f, g** Circulating levels of IPA levels do not correlate with blood neutrophil and lymphocyte abundance in CRC patients. **h** IPA concentration positively correlates with high lymphocyte to monocyte ratio (LMR) in CRC patients. **i** High monocyte count is associated with worse survival in patients with localized CRC. Analysis from the institutional database of the University Hospital of Geneva (HUG). Data are shown as mean ± SD and analyzed using two-tailed unpaired Student's t-test (**c**) and two-tailed Wilcoxon rank-sum test (**d**), Pearson correlation test (**e–h**). Icons were created in *BioRender under the license SABINE, A. (2025)* https://BioRender.com/ *pmy8rs7*. Source data are provided as a Source Data file.

mouse model[89]. Although neither study directly evaluated whether IPA affects bone marrow function, the findings suggest that modulating IPA could be a promising avenue for regulating myelopoiesis in the context of aging or autoimmune diseases.

Our study is not without limitations. The molecular targets and effectors of IPA in myeloid progenitors, as well as its effects on other hematopoietic and metastatic niche components, remain to be identified. The extent to which intestinal mucositis depends on the presence or the initial composition of the microbiota was not assessed in the metastatic model and warrants further investigation. Follow-up study is needed to determine if the magnitude of IPA fluxes upon chemotherapy can be used for prognostic stratification of CRC patients with localized disease.

In conclusion, we demonstrate that standard-of-care chemotherapy-induced intestinal mucositis and microbiota reprogramming are essential for protective chemomemory in CRC. This effect was in part modulated by increased blood IPA concentrations. IPA reduced the ability of bone marrow progenitor cells to produce immature immunosuppressive myeloid cells by driving myeloid cell maturation in the bone marrow. This reduction in Ly6C$^{high}$ monocyte production promoted CD4$^+$ T cell antitumor activity. We further provide evidence that IPA could be used concomitantly with standard of care chemotherapy to treat immune-cold CRC liver metastasis. IPA levels were increased by chemotherapy in a subset of CRC patients, suggesting that individual IPA level changes, alone or in combination with other parameters such as circulating Ly6C$^{high}$ monocytes, could be used to predict risk of relapse after adjuvant chemotherapy.

## Methods
### Human samples
All experimental procedures conducted on clinical human tissue samples adhered to the ethical standards set by the institutional and/or national research committees. These procedures also complied with the principles outlined in the 1964 Helsinki Declaration. All participants in this study signed informed consent. The collection of blood samples during the clinical follow up of patients at the Department of Oncology in the University Hospital of Geneva (HUG) was authorized by the Commission Cantonal d'étique de la recherche de Genève under the protocol 2023-01922. Blood samples were collected before treatment or at the end of treatment infusion and immediately processed for serum extraction. We used anonymized institutional database from the University Hospital of Geneva (HUG) for overall survival and monocytes count analysis. The patients were matched with controls from the SwissChronoFood study by sex and age (CER-VD 2017-00487, clinicaltrials.gov NCT03241121)[73]. For this analysis, we used data from the observational phase of this trial. The use of colorectal cancer cells was reviewed and approved by the ethics committee of the Canton of Vaud and the Centre Hospitalier Universitaire Vaudois (CHUV), under the protocol CHUV_DO_CTE_TRP_0001_2017.

### Animal procedures
All experiments were approved by the Animal Ethics Committee of the Canton of Vaud (VD3588 and VD3961). We used C57BL/6 female mice between 8-14 weeks (18g-24g) from Envigo. Experiments were performed in 8-14 weeks old females (18g-24g), aged matched and cohoused for at least 10 days. Experiments were initiated after at least 7 days acclimatation and cohousing. Chemotherapy cycle consisted of one intraperitoneal (i.p.) injection of oxaliplatin (3.5 mg/kg, #2623, Tocris), followed 1 h later by i.p. injection of 5-fluorouracil (5-FU, 50 mg/kg, # F6627, Sigma-Aldrich) and a single dose of 5-FU (50 mg/kg), the next day. The lesions were liver metastases. Size was not measured, but mouse well-being was monitored through body weight,

activity, grooming, and posture, in accordance with the approved ethical protocol.

### Mouse organoids culture and cell culture
*Apc$^{-/-}$; Kras$^{LSLG12D}$; Tp53$^{-/-}$; Smad4$^{-/-}$*-mCherry (AKPS) and *Apc$^{-/-}$; Kras$^{LSLG12D}$; Tp53$^{-/-}$* (AKP) organoids were previously described[90]. Organoids were cultured in 24 well plates in 40 µl Matrigel dome (Corning; # BD356231 or Cultrex; # BME001) in 600 µl in advanced DMEM/F12 1x medium (Gibco # 12634010) supplemented with 10 mM HEPES (Gibco # 15630-056), 2 mM L-Glutamine (Life Technologies™), 100 mg/mL penicillin/streptomycin (1% PenStrep), B27 (Gibco; #17504-044), N2 (Gibco; # 17502-048). For passaging, the organoids were mechanically dissociated and incubated at 4° for 20 min in cell recovery solution (Corning # 354253). The organoids were incubated in TrypLE Select 1X (Gibco # 12563-029) at 37° for 5 min. The single cell suspension was washed in PBS, resuspended in 40 µl Matrigel and cultured as describe. For organoid expansion, wells were split 1:5. For in vivo experiments, $1 \times 10^5$ cells per dome were plated 2 to 3 days before implantation. MC38-GFP cells were cultured in complete DMEM medium (Gibco, ## 31966-021) supplemented with 1% P/S (Thermo Fisher Scientific, # 15140122) and 10% FBS at 37 °C in 5% CO$_2$.

### Human CRC tumoroids culture and in vitro assays
Colorectal cancer cells were isolated from CRC biopsies obtained from the Centre Hospitalier Universitaire Vaudois (CHUV)[91]. Organoids were established by embedding CRC cells in growth factor-reduced Matrigel (~$2 \times 10^4$ cells per 20 mL Matrigel dome) and cultured in Advanced DMEM/F-12 supplemented with 1× GlutaMAX (Thermo Fisher Scientific, Catalog No. 35050038), 10 mM HEPES (Thermo Fisher Scientific, Catalog No. 15630056), 100 µg ml$^{-1}$ penicillin−streptomycin (Thermo Fisher Scientific, Catalog No. 15140122), 1× B-27 supplement (Thermo Fisher Scientific, Catalog No. 17504001), 1 mM N-acetylcysteine (Sigma-Aldrich, Catalog No. A9165), and 50 µg ml$^{-1}$ Primocin (Invivo-Gen, Catalog No. ant-pm-2). A detailed protocol describing organoid culture can be found elsewhere. A luciferase-ZsGreen reporter was introduced in CRC cells as previously described[92].

For monitoring viability, transduced CRC organoids were dissociated using TrypLE Express (Thermo Fisher Scientific, Catalog No. 12605028), filtered (40 µm), and seeded at $10^4$ cells per well in black-wall round bottom ultra-low attachment spheroid microplates (Corning, Catalog No. 4515). IPA (Sigma # 57400-5 G) or vehicle (DMSO) sera were added at different concentrations. Live-cell luciferase readings were performed by adding 150 µg ml$^{-1}$ luciferine (GOLDBIO, Catalog No. LUCK-500) and measuring luminescence in a Tecan Infinite F500 reader (Tecan).

For imaging experiments, transduced CRC organoids were dissociated as indicated above and seeded in microcavitiy plates (SUN Biosciences, Catalog No. Gri3D-96P-S-24-500) at $3 \times 10^4$ cells per well (~400 cells per microcavity). IPA and portal blood sera were used at the concentrations indicated above. Brightfield and fluorescent imaging was performed using a Nikon Eclipse Ti2 inverted microscope system equipped with a 4×/0.13 NA, 10×/0.30 NA, and 40×/0.3 NA air objectives and a DS-Qi2 camera (Nikon Corporation).

### In vitro IPA organoids assay
To test the effect of IPA on AKPS organoids in vitro, $1 \times 10^4$ AKPS cells were plated in 40 µl Matrigel (Corning; # BD356231) dome in 24 wells plate and cultured in organoids cell culture medium containing 50, 100, 200 µM of IPA (sigma # 57400-5 G) or DMSO as vehicle for three days. At the end of the experiments, 5 pictures per wells were acquired using a stereomicroscope (Leica M205 FA). Organoids were collected in RLT buffer containing beta-mercaptoethanol and RNA was isolated using Qiagen mini kit (RNAsy Qiagen mini kit; #74134).

## Liver metastasis model intrasplenic injection

The hepatic model of CRC metastasis in hemisplenectomized mice was generated as previously described[93]. In brief, AKPS organoids were mechanically dissociated for 20 min with gentle shaking at 4° using cell recovery solution (Corning). MC38-GFP cells were cultured as described above. Cancer cells were washed twice with PBS and resuspended in 40 µl Matrigel (AKPS) and PBS (MC38-GFP) for injection. Mice were anesthetized, the spleen was exteriorized via a small incision in the upper left flank, and $3\text{-}4 \times 10^5$ AKPS tumor organoids or $2.5 \times 10^5$ MC38-GFP cells were injected in the semiseparate spleen parenchyma using insulin syringe. Peritoneal cavity and skin were sutured. Mice received buprenorphine as analgesia. Mice were analyzed according to the timeline indicated in the corresponding figures.

## Microbiota depletion, IPA gavage and depletion of immune cells

Microbiota depletion was carried as described previously[20]. Briefly, mice received 2.5 mg/ml enrofloxacin (Baytril 10%, Bayer) in drinking waters for two weeks followed by two weeks of 0.8 mg/ml amoxicillin l and 0.114 mg/ml acid clavulanic (Co-Amoxi-Mepha, Mepha, Sandoz) in drinking water[20]. Antibiotics and cages were changed every other day.

Mice were gavage daily with IPA (Sigma # 57400-5 G) at 20 mg/kg diluted in sterile PBS, as described[35]. Control mice received PBS.

To deplete CD4+ T cells, mice were i.p. injected with 10 mg/kg of anti-CD4 (Bioxcell; ##BE0003-1) or 10 mg/kg isotype control IgG2b (BioXcell #BE0090) at day 3, 5 and 8 after hepatic metastases implantation.

## Metabolomics

For polar metabolite extraction of tissue samples, 20 mg tissue was mixed with 200 µl acetonitrile:methanol:water (40:40:20) and homogenized with a TissueLyser for 2 min. Samples were centrifuged at 15,000 g for 30 s at 4 °C; the supernatants were transferred to a new tube and used for mass spectrometry analysis. For serum samples, polar metabolites were extracted with 10x volume of room temperature 80% methanol. Metabolite extracts were analysed using flow injection analysis–time-of-flight mass spectrometry on an Agilent 6550 Q-TOF instrument, as described by Fuhrer et al.[94]. Mass spectra were recorded from a mass/charge ratio of 50–1000 in high-resolution negative ionization mode. Ions were annotated by matching their measured mass with reference compounds from the Human Metabolome Database (HMDB 4.0), with a tolerance of 1 mD. Statistical significance was calculated using Student's t-test and corrected for multiple hypothesis testing with the BH method. Statistical analyses were performed in MATLAB R2022b (The MathWorks) using an in-house developed pipeline and using Metaboanalyst to perform Metabolites subclass enrichment analysis[30]. Ion intensities data are provided in Supplementary Data 1–3.

The absolute serum concentrations of 3-indolelactic acid (m/z $204 \rightarrow 142$, RT = 2.6), 3-indolepropionic acid (m/z $188 \rightarrow 59$, RT = 3.5), and tryptophan (m/z $205 \rightarrow 188$, RT = 0.93)) were quantitated at Metabolon facility by LC-MS/MS. Stock and internal standard solutions were prepared by dissolving an appropriate amount of material in ACN: water (1:1). These stock solutions were diluted to the Standard H level and further diluted to each of the other concentration levels. For sample preparation, serum was thawed on ice and vortexed for 10–15 s before aliquoting. For blank, blank-IS, and calibration standards, 0.0500 mL PBS was pipetted into the appropriate wells of a 2 mL 96-well plate. For matrix QC samples, 0.0500 mL QC serum was added, and for study samples, 0.0500 mL serum was dispensed into designated wells. Calibration standards were prepared by adding 0.0200 mL of the corresponding calibration spiking solution. Internal standard working solution (0.0200 mL) was added to calibration standards, blank-IS, QC samples, and study samples. Blank samples received 0.0400 mL ACN/water (1:1), while blank-IS, QC samples, and study samples received 0.0200 mL ACN/water (1:1). Methanol

(0.200 mL) was then added to all wells. Plates were capped, vortexed for 1 min, and centrifuged for 10 min at 4000 rpm. A 0.100 mL aliquot of the resulting supernatant was transferred to a fresh 650 µL 96-well plate, capped, and submitted for LC-MS/MS analysis. SAMPLE ANALYSIS: A 2.0 µL aliquot of supernatant was injected onto an Agilent 1290/Sciex QTrap 5500 LC-MS/MS system equipped with a Waters Acquity BEH C18 column (1.7 µm, 2.1 × 100 mm). Chromatographic conditions were as follows: Column temperature: 50 °C, Autosampler temperature: 4 °C, Mobile Phase A1: 0.05% formic acid in water, Mobile Phase B1: 0.05% formic acid in acetonitrile, Flow rate: 0.550 mL/min, Injection volume: 2.0 µL (adjusted as needed for sensitivity), Needle wash: methanol/water (1:1). The MS/MS parameters were as follow Curtain Gas: 35, Collision Gas: Medium, IonSpray Voltage: −4500 V, Temperature: 500 °C, Gas 1: 70, Gas 2: 70, Scan Mode: Multiple Reaction Monitoring (MRM), negative mode. Further information can be found in Supplementary Data 6.

The peak area of the individual analyte product ions was measured against the peak area of the product ions of the corresponding internal standards. Quantitation was performed using a weighted linear least squares regression generated from fortified calibration standards prepared immediately prior to each run. The LC-MS/MS raw data is collected and processed using SCIEX software Analyst 1.6.3 and processed using SCIEX OS-MQ software.

Alternatively, targeted analysis of selected serum samples was performed at the metabolomic facility of the University of Lausanne. Serum (20 µL) was extracted with ice-cold methanol (180 µL) spiked with internal standards (IS). After centrifugation (4 °C, 21,000 g, 15 min), the supernatants were evaporated to dryness, reconstituted in water (50 µL), vortexed, and sonicated for 3 min. Each extract was then centrifuged again (4 °C, 21,000 g, 15 min) and the supernatant transferred to an LC–MS vial for LC–MS/MS analysis as described below. Calibrators were generated following the same procedure, by the addition of methanol spiked with IS mixture to each calibrator (20 µL), vortexed and transferred to LC-MS vials for the injection.

Extracts were analyzed by Reversed Phase Liquid Chromatography coupled to tandem mass spectrometry (RPLC-MS/MS) on a 6495 triple quadrupole system (QqQ) interfaced with 1290 UHPLC system (Agilent Technologies), in the analytical conditions modified from van der Velpen et al.[95]. Data were acquired in Dynamic Multiple Reaction Monitoring (dMRM) mode with a total cycle time of 400 ms, applying optimized collision energies for each metabolite, using MassHunter (Agilent technologies, version B.07.00). Chromatographic separation was performed on a Acquity HSS T3 (1.8 µm 2.1 mm × 100 mm) column (Waters, Ireland), in positive ionization mode. The mobile phase consisted of A = 5 mM ammonium formate and 0.1% formic acid in H2O and B = 100% methanol. The gradient was as follows: 0 min 0 %B, 2 min 0 %B, 4 min 5 %B, 5 min 10 %B, 7 min 90 % B, 8 min 0 %B, 11 min 0 %B. The flow rate was 300 µL/min, the column temperature 20 °C, and the injection volume 2 µL. ESI source parameters were as follows: dry gas temperature 290 °C, dry gas flow 14 L/min, nebulizer pressure 45 psi, sheath gas temperature 350 °C, sheath gas flow 12 L/min, nozzle voltage 500 V, and capillary voltage +4000 V. Further information can be found in Supplementary Data 6.

Raw LC–MS/MS data were processed using MassHunter Quantitative Analysis 10.0. Absolute quantification of indole metabolites was performed based on calibration curves and the stable isotope-labeled internal standards (IS) which were used to determine the response factor for each metabolite. Linearity of the standard curves was evaluated for each metabolite using a ten-point range; in addition, peak area integration was manually curated and corrected where necessary.

## DNA isolation stool

One fecal pellet per mouse was freshly collected into sterile 1.5 ml Biopure tube (Eppendorf). They were quickly placed on dry ice and stored at −80 until further use. We extracted total bacterial DNA with

QiaAMP Fast DNA stool Mini kit (Qiagen # 51604) according to the manufacturer's instructions. In the final step DNA was eluted in 80ul AE elution buffer.

## RT-qPCR

RNA was isolated as described by manufacturer (RNAsy Qiagen mini kit; #74134). RNA concentration and quality were determined using a 2100 Bioanalyzer (Agilent). First Strand cDNA Synthesis Kit (Roche, #04896866001) was used for cDNA production. qPCR analyzes were done using StepOnePlus (Applied Biosystems), SYBR Green PCR Master Mix (Kapa Biosystems/Bioline) and 1uM primers pairs. Data were normalized to the housekeeping gene 18S, and gene expression was determined using the comparative $C_t$ ($\Delta\Delta C_t$) method. Results are shown as fold change over control samples. To validate microbiota depletion, bacterial load was quantified by qPCR on the 16 s rRNA gene expression in stool as described elsewhere[96]. The average ct value of non-antibiotics treated mice was divided by the Ct value of each mouse and these values were logarithmically converted. Primer list can be found in Supplementary Table 2.

## Assessment of microbiome composition by 16S rRNA sequencing

We extracted total bacterial DNA with QiaAMP Fast DNA stool Mini kit, as described above. V4 amplicons were generated using the KOD One PCR Master Mix (Sigma Aldrich) with universal bacterial primers – 515 F (5'-AATGATACGGCGACCACCGAGATCTACAC-TATGGTAATT-GT-GTGCCA GCMGCCGCGGTAA-3') and 806 R (5'-CAAGCAGAAGACGGCATACGA GAT-NNNNNNNNNNNN-AGTCAGTCAG-CC-GGACTACHVGGGTWTCTA AT), consisting of 5' Illumina adapter/RC of Illumina 3' adapter-(unique 12 bp long Golay barcode represented by 'Ns', exclusive to 3' primer)-pad-linker-primer. Duplicates of each amplicon were mixed, quantified with Qubit dsDNA quantification assay (Invitrogen), and pooled at equimolar concentrations. The final 16S amplicon library was purified using GeneJET PCR purification kit (Thermo Scientific) and sequenced on the MiSeq Illumina platform (Illumina MiSeq V2 500 cycles, 250 PE). Paired-end raw reads were demultiplexed and processed using a custom QIIME2 pipeline[97]. Briefly, sequences were trimmed with cutadapt algorithm, paired ends were merged with vsearch algorithm, quality-filtered using q-score method, and chimera-checked[98,99]. Greengenes2 2022.10 database was used to construct the reference sequence file, on which the Naïve Bayes classifier was trained and then used to perform taxonomic assignment to the species level using representative sequences from each OTU[100]. PICRUSt2 (v. 2.5.2) was used to predict functional abundances within microbiome communities[32]. The OTU sequences and abundances per sample obtained using the QIIME2 pipeline after chimera removal were used as input for KEGG orthology (KO) abundance estimation with PICRUSt2. The ggpicrust2 package (v. 1.7.3) for R was used to convert KO abundance data to KEGG pathway abundance data, for visualization and for differential abundance testing using the implemented ALDEx2 method[101].

## Flow cytometry of liver and blood

Livers were minced and resuspended in DMEM (Gibco, # 31966-021) containing 2.5 mg/ml liberase (Roche, #05401020001) and 10 U DNAse (Sigma-Aldrich, #11284932001). Samples were then digested for 30 min at 37 °C with gentle agitation, filtered through 100 μm strainer, washed three times in DMEM (Gibco, # 31966-021) containing 2% FBS. For immune cell phenotyping, immune cells were separated from other cells by Percoll gradient. The single cells were incubated in 2 ml RBC lysis buffer (Invitrogen #00-433-57) for 5 min, washed three times in PBS and filtered through 70 μm strainer. Single cell suspensions were incubated with anti-CD16/CD32 antibody (#14-0161-82, eBioscience) and stained with surface antibody mix for 20 min at 4 °C.

For blood analysis, blood was collected by intracardiac puncture in EDTA-coated tubes. After centrifugation and a PBS wash step,

cell pellets were incubated in RBC lysis buffer for 5 min. Subsequent blocking and staining steps followed the protocol described above.

The list of surface antibodies used is as followed: anti-CD45 (APC, Biolegend, #103111; PE-Cy7, Thermofisher # 25-0451-82; BV605, Biolegend #103140), anti-CD4 (BUV737, Thermofisher # 367-0041-82; FITC, Biolgends #100509), anti-CD62L (BUV395, BD # 740218), Anti-I-A$^b$/IE (BV786, Biolegend # 107645), anti-Ly6g (BV650, Biolegend # 127641), anti-CD11c (SB605, Biolegend # 63-0114-82), anti-CD19 (Pacific blue, Biolegend #152415; PE, BD # 557399), anti-CD44 (PerCP-Cy5.5, eBio # 45-0441-82), anti-Ly6c (FITC, BD # 553104; PerCP-Cy5.5, EBioscience #45-5932-81), anti-CD11b (PE-Cy5, Thermofisher Invitrogen # 61-0112-82), anti-CD161 (PE, eBio # 12-5941-82), anti-CD3e (APC-Cy7, Biolegend # 100330), anti-CD8a (BV711, Biolegend #100748), anti-F4/80 (Alexa 647, Biolegend #123211), anti-PD-1 (Bv711, Biolegend # 135231), anti-MHCII (BUV785, Thermofisher # 417-5321-81). To exclude dead cells, cells were stained with Fixable Aqua Dead Cell Stain Kit (BV510, Thermofisher #L34957) or 7-AAD-Viability Staining (eBioscience # 00-6993-50). For intracellular cytokine and transcription factor staining, cells were subjected to fixation and permeabilization using the Foxp3 / Transcription Factor Staining Buffer Set (Thermofisher # 00-5523-00) according to the manufacturer's instructions. The list of intracellular antibodies used were as followed: anti-TNF-α (FITC, Biolegend #506304), Anti-IFN-γ (APC, Biolegend #505809), anti-granzyme B (Pacific blue, Biolegend # 515407).

To detect MC38-GFP+ cells in the liver by FACS, the livers were processed as described above but without the gradient separation step. Total liver cells were stained with anti-CD45 (APC, Biolegend, #103111), anti-CD31 (PE-Cy7, Biolegend #102417) and fixable Fixable Aqua Dead Cell Stain Kit (Thermofisher #L34957). Gating strategy to isolate MC38-GFP cells: total cells (FSC-A vs SSC-a), single Cells (FSC-H vs FSC-A), single cells (SSC-A vs SSC-h), alive (CD45 vs BV510 aqua dead), GFP positive cells (SSC-A vs GFP). Samples were run on an LSR-II and Fortessa 1 and analyzed with FlowJo 10.

## Analysis of hematopoietic stem and progenitor cells by flow cytometry and cell sorting

After isolating from femur, tibia and pelvis of C57Bl6 mice, cells suspension of bone marrow was filtered through 70 μm cell strainer and erythroid cells were eliminated by incubation with red blood cells lysis buffer (BioLegend). For analysis of hematopoietic stem and progenitor cell (HSPC) compartments, cell suspensions were then stained with antibodies for progenitor and stem cell compartments as the following: HSCs: Lineage$^-$cKit$^+$Sca1$^+$(LKS)CD150$^+$CD48$^-$, Multipotent progenitors (MPPs): LKSCD150$^-$, Common lymphoid progenitor (CLP): CD127$^+$cKit$^{low}$Sca1$^{low}$, Common myeloid progenitors (CMPs): Lineage$^-$cKit$^+$Sca1$^-$(KLS$^-$)CD16/32$^{low}$CD34$^+$, Granulocyte–macrophage progenitors (GMPs): KLS$^-$CD16/32$^{high}$CD34$^+$ and Megakaryocyte-erythroid progenitors (MEPs): KLS$^-$CD16/32$^{low}$CD34$^-$. Cells were then washed with FACS buffer and analyzed by flow cytometry on BD LSR II. For cell sorting, lineage positive cells in the cell suspensions were first removed with a magnetic lineage depletion kit (BD biosciences). Cell suspensions with depleted lineage positive cells were then stained with specific antibodies for LKS compartment and sorted on BD FACS Aria III and Aria IIu. Isolation and staining were performed in ice-cold PBS supplemented with 1 mM EDTA.

## CMP adoptive transfer

For CMP adoptive transfer experiment, we followed existing protocols[71]. 20,000 or 40,000 CMP (Lineage$^-$cKit$^+$Sca1$^-$ (KLS$^-$)CD16/32$^{low}$CD34$^+$) were FACS sorted from the bone marrow of CD45.1 mouse bearing metastasis which received daily treatment with 20 mg/kg IPA or PBS. CMPs were injected in the tail vein of CD45.2 recipient mice. The bone marrow and blood were harvested 7 days later for CD45.1 cell fate tracking by flow cytometry.

Cell gating for sorting: Lymphocytes (FSC-A vs SSC-A), Single Cells (FSC-H vs FSC-A), Live cells (Zombie NIR$^-$ and SSC-A), Lin- (Lin- and SSC-A), Lineage$^-$Kit$^+$Sca1$^-$, Common myeloid progenitors (CMPs): Lineage$^-$Kit$^+$Sca1$^-$(KLS$^-$)CD16/32$^{low}$CD34$^+$.

Cell gating for analysis: Lymphocytes (FSC-A vs SSC-A), Single Cells (FSC-H vs FSC-A), alive (Aqua blue and FSC-A), CD45.1 (FSC-A and CD45.1), CD11b$^+$ (CD11b$^+$), Monocytes (Ly6c$^+$ Ly6g- gated in CD11b$^+$), neutrophils (Ly6c$^{low}$ Ly6g$^+$ gated in CD11b$^+$), Macrophages (Ly6c$^-$Ly6g$^-$F4/80$^+$ gated in CD11b$^+$).

## CFU assay
Colony-forming unit (CFU) assay with murine bone marrow cells was carried out using MethoCult GF M3434 (STEMCELL Technologies), formulated to support growth of multi-potential granulocyte, erythroid, macrophage, megakaryocyte progenitor cells, according to the manufacturer's instructions. For in vivo experiment, $2 \times 10^4$ total bone marrow cells from each mouse treated with IPA or PBS were plated in duplicate. Colonies were counted 7 days after plating using Stem Vision software (STEMCELL Technologies).

## T lymphocyte assay and coculture assay
CD4 and CD8 T cells were isolated from the spleen and lymph nodes of wild type mice following manufacturer instructions using CD4 and CD8 mouse isolation kits (MILTENYI BIOTEC INC # 130-104-454 and # 130-104-075,). T lymphocytes were labeled with carboxyfluorescein (CFSE, Thermofisher # C34570) as described by manufacturer. $1 \times 10^5$ CD4 or CD8 cells were stimulated with 2 μg/ml plate coated CD3 (Biolegend #100340) and 3 μg/ml soluble anti-CD28 (Biolegend #102116) resuspended in RPMI media containing 10%FBS, 1%PenStrep, and β-mercaptoethanol. In IPA exposure experiment, CD4 and CD8 cells were exposed to DMSO (control), 10 μM, 50 μM and 500 μM of IPA. In the experiment assessing TCF1 modulation by IPA, Il-2 (50 U/ml) was added to the culture. For coculture experiments, CD4 cells were prepared as above. Ly6C$^{high}$ monocytes were FACS sorted from the liver and bone marrow of mice and plated with CD4 cells at a ratio of 1:4. In all experiments, cells were stained with 7-AAD viability dye after 72 or 96 h culture and proliferation was measured as dilution in CFSE label on an LSR-II.

## Immunosuppressive myeloid cell in vitro generation
Bone marrow cells were isolated as described[102]. $1 \times 10^6$ bone marrow cells were plated in Immunosuppressive myeloid cell-generating media consisting of RPMI medium (Thermofisher # 11875093) supplemented with 10% FBS, 1% PenStrep, 1% glutamine, 20 ng/ml GM-CSF (Peprotech # 315-03), 20 ng/ml IL-6 (Peprotech # 216-16). 0.1% DMSO (control) or 10 μM or 50 μM IPA were added to the culture. The media was changed every two days and the number of Ly6C$^{high}$ monocytes and neutrophils was analysed after 5 days on an LSR-II.

## RNA sequencing
RNA-seq libraries were prepared from 25 ng of total RNA with the Illumina Stranded mRNA Prep reagents (Illumina) using a unique dual indexing strategy, and following the official protocol automated on a Sciclone liquid handling robot (PerkinElmer). Libraries were quantified by a fluorometric method (QubIT, Life Technologies) and their quality assessed on a Fragment Analyzer (Agilent Technologies). Libraries were sequenced at the Lausanne Genomic Technologies Facility on an AVITI Sequencing Instrument (Element Biosciences). Fastq generation was done using Base2Fastq (v. 1.7, Element Biosciences). Sequencing reads were pre-processed and aligned to the reference genome using the RNA-seq pipeline of the bcbio-nextgen tool (v. 1.2.4). The pipeline involved the following steps. Sequencing quality was assessed with FastQC (v. 0.11.8). The reads were trimmed for quality and adapter using Atropos (v. 1.1.28)[103]. Trimmed reads were aligned to the *Mus musculus* genome (mm10) using hisat2 (v. 2.2.1). Numbers of reads were summarized at the gene level per sample using featureCounts (v. 2.0.1) and the Ensembl GRCm38.92 release of the mouse transcriptome annotation. Samtools (v. 1.9) was used for alignment file manipulation, and alignment quality was assessed with MultiQC (v. 1.9) and qualimap (v. 2.2.2 d)[104–107]. Raw counts were imported into R (v. 4.3.0). Genes were filtered to only retain the ones expressed at more than 0.5 count per million (cpm) in at least 2 samples ($n = 14970$ retained genes). Differential gene expression analysis between activated 10 and control samples was performed using the DESeq2 package (v. 1.14.1) and extracting the results using the results function with argument $\alpha = 0.05$[108]. Genes with a p-value $< 0.05$ after Benjamini-Hochberg adjustment were considered as significantly differentially expressed between activated 10 vs control.

Gene set enrichment analysis (GSEA) was performed using the clusterProfiler package (v. 4.8.3) for R[109]. We performed GSEA against the Gene Ontology Biological Process collection with the gseGO function. With the GSEA function, we also tested for enrichment of a combined MDSC signature extracted from Table S5 of Alshetaiwi et al. 2020, and an AHR agonist signature extracted from Table S3 of Goudot et al. 2017[49,110]. The parameters for both the gseGO and GSEA functions were: eps = 0, seed=T and a seed=1234. We inferred transcription factor activity in each sample using the SCENIC package (v. 1.3.1) for R (v. 4.1.2)[111]. The SCENIC package made use of functions from the RcisTarget (v. 1.14.0), GENIE3 (v. 1.16.0), and AUCell (v. 1.16.0) packages and the cisTarget databases (mm10_refseq-r80_10kb_up_and_down_tss.mc9nr.feather and mm10_refseq-r80_500bp_up_and_100bp_down_tss.mc9nr.feather). The AUC score obtained for each regulon in each sample was compared in activated 10 vs control using a T-test, followed by p-value adjustment with the Benjamini-Hochberg procedure. A heatmap of the scaled AUC score per sample of the significant regulons was generated using the ComplexHeatmap package (v. 2.16.0)[112].

## Imaging mass cytometry
Slides were stained following existing protocol (Supplementary Data 4)[113]. Data were analyzed following the "Analysis workflow for IMC data" at https://bodenmillergroup.github.io/IMCDataAnalysis/index.html, without batch effect and spillover correction steps. T cells were split into CD4$^+$, CD8$^+$, and Tregs (CD4$^+$ Foxp3$^+$), lymphocytes into macrophages (CD11b$^+$ F4/80), myeloid cells (CD11b$^+$ F4/80- Ly6g-), and neutrophil-like (CD11b$^+$ Ly6G$^+$), endothelial cells into CD31$^+$ and CD31$^+$ aSMA$^+$ groups, and hepatocytes into periportal (CDH1$^+$), pericentral (GS$^+$), and intermediate (negative for both markers). Cells were additionally sorted into negative/positive groups based on marker levels for Tbet. Annotated cell quantities were then used for differential abundance analysis as described in chapter 6 of "Multi-Sample Single-Cell Analyses with Bioconductor" at https://bioconductor.org/books/3.13/OSCA.multisample/ using R package edgeR[114] v. 3.42.2.In order to assess differential marker expression in cell types between conditions, differential state analysis was performed mimicking diffcyt[115] *diffcyt-DS-limma* function, using R package limma[116] v. 3.56.1. Spatial organization of identified cell types was assessed using R package lisaClust v. 1.8.2. Resulting regions contain mostly tumor cells (Tumor), inflammatory cells (Inflammatory), larger proportion of myeloid cells (Myeloid), or represent liver regions with most cells being hepatocytes (periportal – PP, intermediate – IM, pericentral – PC). Additionally, cells were tested for attraction/avoidance using imcRtools function *testInteractions*. The raw and processed data and the analysis scripts are available at https://doi.org/10.5281/zenodo.17574413.

## Immunofluorescent staining
Tissues were harvested, fixed in 4% PFA overnight, embedded into paraffin or OCT (Sakura). Paraffin sections were deparaffinized and subjected to heat-induced epitope retrieval using Citrate- (Vector Laboratories; H-3301-250) or Tris-based (Vector laboratories; H-3301-250) buffers. Cryosections were thawed for 20 min at room

temperature, fixed in PFA 4% for 5 min, washed with PBS and permeabilized with 0.3% Triton-X-100 in PBS, followed by 30 min blocking, overnight incubation with primary antibody and secondary antibody staining for 1 h. Slides were mounted in Fluoromount-G mounting medium with DAPI (Invitrogen, #00-4959-52).

## Images quantification and analysis

Sections of entire liver were scanned using a NanoZoomer S60 slide scanner (Hamamatsu) with a 20x objective. Images were analyzed with NDP.view2 (Hamamatsu), and open-source software Qupath[117]. For metastatic burden quantification, we isolated liver lobes, manually annotated each metastatic lesions defined as mCherry⁺, E-cadherin⁺ or CD44v6⁺ for AKPS or as GFP⁺ for MC38-GFP cells and calculated the total metastatic area in $\mu m^2$. The number of lesions or their total area were normalized to the total liver area. For quantification of proliferating cells using PH3 staining, CD44v6 area (specific for AKPS metastasis) were isolated using pixel classifier in QuPath, converted into annotation and the number of PH3⁺ cells within the metastatic lesion were quantified using object classifier (nucleus mean measurement). The average of PH3⁺ cells per mouse was then calculated and used for statistical analysis. For quantification of necrosis within metastatic lesion, 4 to 5 metastatic lesions were isolated per mouse and necrotic area, defined as thinning of the epithelium, loss of cell architecture, nuclei and presence of cellular debris, was manually drawn. Necrotic area was then normalized to metastatic area and control mean.

## Figure preparation

Icons were created in BioRender under the license SABINE, A. *(2025)* https://BioRender.com/ *pmy8rs7*. Icons and graphs were assembled and modified in Affinity.

## Statistical analysis

All data are shown as mean ± standard deviation. Data analyses were carried out with GraphPad Prism 10 (GraphPad) for MacOS. Two groups were analyzed using a two-tailed unpaired Student's t-test, or in cases where experiments comprised more than two groups, a one-way ANOVA with Tukey post hoc test was employed. In vitro experiment, exposing T cells to IPA and Ly6C^high monocytes generating experiments were analyzed using a multiple permutated one-way ANOVA. The numbers of biological replicates are stated in the figure legends. P-value < 0.05 were considered statistically significant. The statistical methods for each complex data analysis, such as RNA-seq, metabolomics, and spatial analysis, are detailed in their respective sections.

## Data availability

The RNA-seq and 16 rRNA seq data generated in this study have been deposited in the GEO database under accession GSE282931 and GSE29690, respectively. The metabolomics data were deposited in MetaboLights (MTBLS13162). The raw and processed data and the analysis scripts for CyTOF are available at https://doi.org/10.5281/zenodo.17574413. The remaining data are available within the article and Supplementary Information. The source numbers for graphs are available in the Source Data file whenever possible. Source data are provided with this paper.

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

## Acknowledgements

We are grateful to all patients who generously donated blood for this study. We thank Nicole Jaquenoud, Sebastien Bugeia, Garance Gut-knecht and Jessica Rebeaud for patients' samples handling and patients' recruitments. We thank Mikael Pittet, Michele De Palma and Yahya Mohammadzadeh for useful discussions. We thank Astrid Huchet, Aimad Ourahmoune and Valentin Rochat for the analysis of the CRC HUG patient database. We gratefully acknowledge the help of Valérie Dutoit for human samples processing organization. We thank Amélie Sabine, Kelly De Korodi, Katerina Pandeva and Celine Beauverd for lab and mouse colony management, technical assistance with some experiments and help with figure preparation. We express our gratitude to the Animal, Cellular Imaging, Genomic Technologies, Mouse Pathology and Flow Cytometry and the Metabolomics and Lipidomics Facilities at UNIL and Imaging Mass Cytometry at University of Bern. We thank Prof. Andreas Moor and Dr. Costanza Borrelli for sharing AKPS organoids and Prof. Lubor Borsig for MC38-GFP cells. This work was supported by the Swiss National Science Foundation and Swiss Cancer Research (MD-PhD 5089-06-2020 to L.B., SNSF 10003598 to C.P., SNSF 316030_183501 to D.S.), Bryn Turner-Samuels Foundation (to L.B.), Joseph and Lina Spicher Foundation (to L.B.), KFS-4895-08-2019 (to T.V.P.) and an ISREC Tandem grant (to T.V.P. and T.K.). The SwissChronoFood trial is supported by grants from the Swiss National Science Foundation (SNSF, PZ00P3-167826, 32003B-212559 to T.H.C.). S.B. is supported by an Eccellenza Professorial Fellowship (PCEFP3_187018) and a Starting Grant (TMSGI3_211235) by the Swiss National Science Foundation. A.C. is supported by a PhD Salary Award by the iGE3 Genomics Platform of the University of Geneva.

## Author contributions

T.V.P. and L.B. conceived the study. L.B., L.F.L.M., S.D., Y.H.C., and A.C. performed experiments and analyzed the data. I.R., J.B.L., and J.K. performed experiments. L.B. prepared the figures. T.W. and A.C. performed microbial bioinformatics analysis. T.W. analyzed RNAseq data. T.Y. analyzed CYTOF imaging. C.P. and T.H.C. are principal investigators of the SwissChronoFood trial (ClinicalTrials.gov registry no. NCT03241121) and contributed to the metabolomic analysis. T.K. is responsible for CRCCT patients' recruitment. CYTOF imaging analysis was supervised by D.S. Microbial sequencing was supervised by S.B. M.P.L. supervised human tumoroid experiments. Untargeted metabolomics analyses were supervised by N.Z. Bone marrow analyses were supervised by N.V. L.B. and T.V.P. wrote the manuscript. All authors read and reviewed the manuscript.

## Competing interests

The authors declare no competing interests.

## Additional information

¹Department of Fundamental Oncology, University of Lausanne and Ludwig Institute for Cancer Research Lausanne, Lausanne, Switzerland. ²Laboratory of Stem Cell Bioengineering, Institute of Bioengineering, School of Life Sciences and School of Engineering, Ecole Polytechnique Fédérale de Lausanne, Lausanne, Switzerland. ³Institute of Molecular Systems Biology, Department of Biology, ETH Zürich, Zürich, Switzerland. ⁴Department of Pathology and Immunology, Faculty of Medicine, University of Geneva, Geneva, Switzerland. ⁵Geneva Centre for Inflammation Research, Faculty of Medicine, University of Geneva, Geneva, Switzerland. ⁶Department of Visceral Surgery and Medicine, Inselspital, Bern University Hospital, University of Bern, Bern, Switzerland. ⁷Department for BioMedical Research, University of Bern, Bern, Switzerland. ⁸SIB Swiss Institute of Bioinformatics, Lausanne, Switzerland. ⁹Service of Neurology, Department of Clinical Neurosciences, Lausanne University Hospital (CHUV) and University of Lausanne, Lausanne, Switzerland. ¹⁰Service of Endocrinology, Diabetes and Metabolism, Department of Medicine, Geneva University Hospitals, Geneva, Switzerland. ¹¹Diabetes Centre, Faculty of Medicine, University of Geneva, Geneva, Switzerland. ¹²Institute of Human Biology (IHB), Roche Pharma Research and Early Development (pRED), Roche Innovation Center Basel, F. Hoffmann-La Roche Ltd., Basel, Switzerland. ¹³Center for Translational Research in Oncology-Hematology, Department of Oncology, University Hospital of Geneva, Geneva, Switzerland. ¹⁴Department of Oncology, Geneva University Hospital, Geneva, Switzerland. ¹⁵Swiss Institute for Experimental Cancer Research (ISREC), School of Life Sciences, EPFL, Lausanne, Switzerland. ¹⁶Present address: Department of Ophthalmology, Samsung Medical Center, Sungkyunkwan University School of Medicine, Seoul, Republic of Korea. ✉e-mail: tatiana.petrova@unil.ch

