## [Transparent Peer Review file · Nature Communications]

Chemotherapy-driven intestinal dysbiosis and indole-3-propionic acid rewire myelopoiesis to promote a metastasis-refractory state

Corresponding Author: Professor Tatiana Petrova

Version 0:

Reviewer comments:

Reviewer #1

(Remarks to the Author)

This manuscript addresses the role of the gut microbiome in chemotherapy induced reduction in colorectal cancer metastasis to the liver. The authors posit that chemotherapy induced intestinal mucositis disrupts the gut epithelial cells resulting in reduced absorptive capacity, altering nutrient availability and causing dysbiosis. There are several studies in the literature that support this process. The novel aspects of this study is the demonstration of the expansion of tyryptophan-metabolizing bacteria that increases levels of IPA which drives myeloid cell maturation to macrophages resulting in reduced Ly6Chigh monocyte production. The authors show that this further promotes CD4 T cell activity thus reducing tumor progression in the liver.

This is a well written manuscript with appropriate methodology. The studies are thorough and the conclusions follow from the data. I have minor suggestions and issues that can be addressed.

- 1) It would be helpful to have a final schematic illustration of the mechanism by which "chemomemory" is thought to occur.
- 2) It would be useful to get more details on the extent of the GI toxicity other than histological measurements. What is the inflammation state in the model among the various cohorts.
- 3) Are there data on the changes in the immune response in the lamina propria?
- 4) It is unclear why intestinal damage is determined after 3 days post FO in naïve mice but 5 days post FO in treated mice (Figs 1A and B)
- 5) Fig 8: What are the monocyte counts for T2? While there is clear shift in IPA the monocyte counts are presented as a combination of the T1 and T2 in 8E..

Reviewer #2

(Remarks to the Author)

The manuscript demonstrates that standard CRC chemotherapy with 5-fluorouracil and oxaliplatin induces intestinal mucositis, which alters the gut microbiota composition, leading to increased levels of the bacterial metabolite indole-3-propionic acid (IPA). Elevated IPA reprograms bone marrow myelopoiesis, reducing immunosuppressive Ly6Chigh monocytes and promoting anti-tumor CD4+ T cell responses. This effect, termed "chemomemory" by the authors, reduces colorectal cancer liver metastasis in mice even after chemotherapy has cleared. IPA supplementation alone can mimic this protective effect, suggesting that it could be used to enhance the efficacy of chemotherapy. Analysis of 14 CRC patients revealed that chemotherapy increases serum IPA levels in some individuals, with IPA inversely correlating with circulating monocytes, supporting its clinical relevance.

The manuscript contains important data on a timely aspect of CRC and metastases. I would encourage the authors to clarify some key points and temper rather speculative interpretations before publication.

Major
Results

In the section “Anti-metastatic chemomemory is microbiota-dependent”: Is mucositis itself microbiota-dependent in their model? Can the authors assess this from their metastasis experiments? While this may not be central to the paper’s claims, it could help clarify the proposed mechanism.

“We tested the effects of high physiological (10 μ M) and supraphysiological (50, 100 and 200 μ M) IPA concentrations (43, 44). Only non-physiological concentration of 100 μ M IPA inhibited growth of patient-derived MSS tumoroids” – How are “physiological” and “supraphysiological” concentrations defined? What evidence supports 10 μ M as physiological in the tumor microenvironment?

“IPA was unable to suppress metastatic growth or tumor cell proliferation in NSG mice (Fig. 4L-O). These results show that in vivo IPA anti-metastatic activity is primarily driven through the immune system.” – I recommend more cautious wording here, as NSG mice differ from immunocompetent models on multiple levels.

Related to Figure 7A-E: Did IPA treatment have any effect on overall tumor size? If not, can the authors speculate why, and could they show the data?

“CRC patients present with lower IPA level as compared to the healthy population, suggesting a role for IPA in disease control and that IPA-producing bacteria are preferentially targeted by antibiotics used to treat these patients.” This interpretation appears highly speculative. Even if the observation holds in larger cohorts, reduced IPA levels could have multiple causes. I suggest a more careful phrasing.

Discussion

5FU/Oxaliplatin is the standard adjuvant therapy for stage III CRC, yet many patients still develop metastases. Can the authors reflect on why the mechanisms they uncover might fail in a significant proportion of patients?

Methods

It appears from the main Methods section that only GraphPad was used for statistical analyses. Is this sufficient for the more complex data, such as proteomics and spatial analyses?

Minor

Introduction

Consider rewording: stage I & II CRC are still “invasive”; consider removing “Non-invasive.”

“antibiotics prior or during chemotherapy treatment significantly reduced disease-free survival in CRC patients (7), suggesting that gut microbiota confers protection against metastatic recurrence.” – Given the potential for many confounding factors, I suggest more cautious phrasing. For example:

“...in CRC patients. However, how altered microbiota, for example due to antibiotics, and tumor progression are mechanistically linked remains incompletely understood.”

Results

“indicating that the metastasis refractory state is long-lasting (Fig. 1L-N; Fig. S1JK). In all, we demonstrate that FO induced mucositis and systemic effect maintain a lasting anti-tumor effect.” The authors use “long-lasting” for a period of ten days. Would “lasting” suffice?

“...suppressed liver metastasis growth to levels similar to those observed in FO-treated, microbiota-competent mice (Fig. 4IJ; Fig. 1H-I).” – As the authors did not measure growth dynamically, it would be more accurate to use “liver metastasis size” or “burden.”

Methods

The Supplementary Methods should be clearly referenced, or, as is more typical for the journal, all methods should be included in the main text.

Reviewer #3

(Remarks to the Author)

The manuscript by Bersier et al. proposes a mechanism how chemotherapy effect systemic immune memory by regulating intestinal microbiota and microbiota-dependent metabolite IPA, which in turn modulates myelopoiesis and subsequent T cell activation suppressing liver metastasis development. While the role of IPA in regulation of T cell immune responses in anti-cancer treatment was previously suggested, this work provides novel insights into the indirect, monocyte-dependent control of anti-tumor T cell responses in liver metastases. The manuscript provides strong experimental evidence supporting proposed mechanisms and use various loss- of- function and gain -of-function approaches and multiple metastasis models to support conclusions.

Overall, this is an exciting study. However, some additional clarifications can further strengthen presented work.

1. What would be the explanation on why antibiotic treatment does not suppress metastasis development (Fig 2)?
2. Metastasis models are relatively short. Is this effect of chemotherapy on microbiota and subsequent immune system rewiring long lasting? It would be important to discuss it in the manuscript.
3. Can authors clarified how would depletion of circulating monocytes affect chemotherapy /IPA mediated T cell immune responses in metastases?
4. Fig 6 E. The gates defining neutrophils need to be corrected. Typically neutrophils are defined as Ly6G^{hi}, Ly6C⁺, unlike presented on this figure.

Reviewer #4

(Remarks to the Author)

Bersier et al. analyzed how chemotherapy-induced normal tissue injury can influence individual susceptibility to metastasis, specifically in colorectal cancer (CRC). The authors showed that chemotherapy indirectly prevents liver metastases by inducing a long-lasting systemic "chemomemory." This effect was mediated by intestinal mucositis, which altered gut nutrient availability and promoted the expansion of tryptophan-metabolizing bacteria, leading to increased production of the microbial metabolite indole-3-propionic acid (IPA). They further showed that IPA reshaped bone marrow myelopoiesis, resulting in reduced numbers of immunosuppressive Ly6C-high monocytes in the circulation and liver. This reduction facilitated the accumulation of CD4⁺ Th1-like T cells and enhanced colocalization of CD8⁺ cytotoxic T cells, ultimately strengthening anti-tumor immunity.

General Remark 1: The authors have went to great lengths to describe the beneficial role of IPA in reinforcing the chemomemory in the context of 5-FU and Oxali treated CRC-bearing animals. The results are in this regard convincing, however they are not novel as this has been reported by other scientists recently (https://ascopubs.org/doi/10.1200/JCO.2024.42.16_suppl.e15194; PMID: 38490195).

General Remark 2: Despite very large animal experiment, the mechanism on how IPA actually works in the cell (eventual receptor or intracellular target) remain unknown.

General Remark 3: The metabolomics analysis has not been well detailed, targeted approach needs the specific information on which transitions have been used, retention times, etc. More detail is needed if these results are to be reproduced in the future.

Reviewer #5

(Remarks to the Author)

Version 1:

Reviewer comments:

Reviewer #1

(Remarks to the Author)

The authors have responded appropriately to my comments. I have no further comments.

Reviewer #2

(Remarks to the Author)

Thank you for your thorough revision of the manuscript. All points raised in my previous review were addressed, and I do not have any further comments.

Reviewer #3

(Remarks to the Author)

Authors successfully addressed previously raised concerns and revised the manuscript accordingly. I have no further questions.

Reviewer #4

(Remarks to the Author)

Bersier et al. have submitted a revised manuscript that addresses some of the reviewers' questions and provides additional clarity on others; however, several issues remain unresolved. The primary concern—the lack of mechanistic insight into the

action of IPA in myeloid cells—remains pertinent. This is particularly important given that the present data appear to conflict with previous findings (PMID: 38490195), where IPA targets different cells via a distinct mechanism. While this does not invalidate the authors' findings, it highlights the need for more definitive mechanistic understanding, especially in light of multiple reports demonstrating the beneficial role of IPA in cancer therapy.

A second major concern pertains to the metabolomics analyses presented throughout the manuscript, including key figures (Figs. 2, 3, and 8), particularly those regarding IPA concentrations in patients. Based on the additional details provided, it is evident that the metabolomics analyses were conducted following the methodology of Fuhrer et al., 2011 (as per authors information), which no longer reflects current standards. Today, it is considered unacceptable to quantify or identify metabolites—as in Figure 3F (untargeted)—without the use of liquid chromatography (retention times are unavailable) and MS² validation. Direct injection and MS¹-based quantification may be acceptable in limited in vitro screening experiments with defined metabolite pools but not in complex biological matrices such as serum or cellular extracts. Similarly, the targeted quantification presented in Figure 3E is problematic. Using a single transition, as now described in the Materials and Methods section, is highly unconventional and does not meet current standards for reliable targeted metabolomics.

Finally, it is unclear why the authors have only partially addressed reviewers' requests for detailed MS methodology—a critical component of the study. The description provided (“...was injected onto an Agilent 1290/Sciex QTrap 5500 LC-MS/MS system equipped with a C18 reversed-phase ... column”) lacks essential details regarding the column type, dimensions, part number, gradient conditions, and solvent composition. Reporting retention times without providing these key parameters limits reproducibility and transparency.

In conclusion, the metabolomics analyses must be performed according to current state-of-the-art standards. Complete and detailed reporting in the Materials and Methods section is essential to ensure reproducibility and adherence to international guidelines.

Reviewer #5

(Remarks to the Author)

Reviewer #6

(Remarks to the Author)

We thank the Reviewers for their positive feedback and thoughtful evaluation of our manuscript. Following the constructive review, we revised the manuscript as suggested and included new data to address reviewer's concerns.

Comments are in blue, and our responses are written in black.

POINT BY POINT RESPONSES TO REVIEWERS' COMMENTS

Reviewer #1 (Remarks to the Author):

This manuscript addresses the role of the gut microbiome in chemotherapy induced reduction in colorectal cancer metastasis to the liver. The authors posit that chemotherapy induced intestinal mucositis disrupts the gut epithelial cells resulting in reduced absorptive capacity, altering nutrient availability and causing dysbiosis. There are several studies in the literature that support this process. The novel aspects of this study is the demonstration of the expansion of tyryptophan-metabolizing bacteria that increases levels of IPA which drives myeloid cell maturation to macrophages resulting in reduced Ly6chigh monocyte production. The authors show that this further promotes CD4 T cell activity thus reducing tumor progression in the liver. This is a well written manuscript with appropriate methodology. The studies are thorough and the conclusions follow from the data. I have minor suggestions and issues that can be addressed.

We thank reviewer 1 for their comment, that helped clarify and strengthen our manuscript.

1) It would be helpful to have a final schematic illustration of the mechanism by which “chemomemory” is thought to occur.

Thank you for useful suggestion. We have now added a final schematic figure to illustrate the mechanisms of chemomemory” (Figure 1 for reviewers and page 16, lines 433-438 and Figure 6M in the manuscript).

Figure 1 for reviewers. IPA role in the bone marrow myeloid compartment and liver metastatic microenvironment. In the absence of IPA, tumor induced Ly6c^{high}CCR2⁺monocytes are released from bone marrow in the circulation and recruited to the tumor microenvironment, where they suppress antitumor immunity. IPA promotes maturation of these monocytes, decreasing their release in circulation. This in turn relieves immunosuppression and enhances antitumor immunity mediated by CD4⁺Th1 cells in metastatic niche, driving IPA-mediated chemomemory.

2) It would be useful to get more details on the extent of the GI toxicity other than histological measurements. What is the inflammation state in the model among the various cohorts.

The reviewer raises a valid point. In Figure 1, we demonstrated histological signs of mucositis, including reduced villus length and crypt depth. Additionally, our metabolomics analysis of the ileum at 5 days post FO revealed a decrease in citrulline levels, an established marker of enterocyte mass (Figure 2 for reviewers 2H, Fig. 2H) (3, 4). Please refer to page 7, lines 188-190 in the manuscript: “In addition, FO-treated ileum contained less citrulline (Fig. 2H, Table S1), a metabolite mostly produced by enterocytes and used as a marker of enterocyte mass(3).” Consistent with inflammation, we also observed increase in other arachidonic acid associated catabolites (5, 6)(page 7, lines 186-188 and Fig. 2H in the manuscript).

Figure 2 for reviewers (from figure 2). FO increase metabolite associated with intestinal damage and inflammation. (H) Hierarchical clustering heatmap showing metabolite abundance changes in control and FO treated ileum ($\log_{2}FC > 0.2$; $FDR < 0.05$). $n = 5$ per condition run in duplicate

In addition, we now analyzed expression levels of inflammatory cytokines in ileum of control or FO treated mice. This analysis demonstrates increase expression of *Tnf α* and *Il1b*, two additional markers of intestinal inflammation (Figure 3 for reviewers. 3A).

We have added these data in **Supplementary Fig. 1J** and modified the text accordingly on **page 5, lines 122-123 in the revised manuscript.**

3) Are there data on the changes in the immune response in the lamina propria?

To further assess immune response in the lamina propria, we performed IHC on sections of mouse intestine at day 3 and day 5 post FO or vehicle treatment (control). Histological analysis showed that FO treatment significantly increased infiltration of MPO+ neutrophils, a known marker of mucositis, while no changes in CD4+ or CD8+ lymphocyte were found at both time

points (**Figure 3 for reviewers. 3B-I**). We have added these data in **Supplementary Fig. 1K-R** and modified the text accordingly on **page 5, lines 123-127**.

Thus, FO treatment, in addition to histological signs of intestinal damage, such as decreased villus and crypt length, also leads increased lamina propria inflammation and expression of inflammatory cytokines. All these new data are now included in **Supplementary Figure 1**.

Figure 3 for reviewers. FO induces intestinal inflammation. (A) qPCR analysis of *Tnfa* and *Il1b* in the ileum of mice 5 days after treatment with control (PBS) and FO. Data are normalized to housekeeping gene *18S*. **(B-C)** MPO+ cells infiltrate the ileum of FO treated mice 3 days post-treatment. Images show MPO (green) and DAPI (blue). Scale bars: 100µm. Quantification MPO+ cells normalized to area. n=4 per groups. **(D-E)** MPO+ cells infiltrate the ileum of FO treated mice 5 days post-treatment. Images show MPO (green) and DAPI (blue). Scale bars: 100µm. Quantification MPO+ cells normalized to area. n=3 per groups. **(F-G)** CD4+ and CD8+ cell infiltration are similar 3 days post-treatment. Images show CD4 (green), CD8 (magenta) and DAPI (blue). Quantification CD4+ and CD8+ cells normalized to area and control mean. n=4 per groups. Scale bars: 100µm **(H-I)** CD4+ and CD8+ cell infiltration is similar 5 days post-treatment. Images show CD4 (green), CD8 (magenta) and DAPI (blue). Quantification CD4+ and CD8+ cells normalized to area and control mean. n=3 per groups. Scale bars: 100µm. Data are shown as mean ± SD and analyzed using two-tailed unpaired Student's t-test (**A, C, E, G, I**).

4) It is unclear why intestinal damage is determined after 3 days post FO in naïve mice but 5 days post FO in treated mice (Figs 1A and B)

This may be a misunderstanding: on Figure 1A and B, the data compare the intestine 3 days after control treatment (PBS, left panel; note that “control” has been added to the figure legend) and 5-FU+oxaliplatin (FO, right panel). To clarify the experimental setup, we have revised data

presentation as shown below (**Fig. 4 for reviewers; Fig. 1A-B in the manuscript**). We hope these changes help improve the clarity of the information.

Figure 4 for reviewers. FO induces intestinal damages. (A) Scheme of the experiment. Wild-type mice received 5-FU+oxaliplatin (FO) or PBS (control); intestines were harvested 3 days later. (B) FO reduced proliferation of intestinal crypt cells 3 days after treatment. KI67 (magenta) and DAPI (grey). Scale bars: 100µm.

5) Fig 8: What are the monocyte counts for T2? While there is clear shift in IPA the monocyte counts are presented as a combination of the T1 and T2 in 8E.

We thank the reviewer for this question. Monocyte levels following chemotherapy vary among CRC patients. Notably, most patients who exhibited a substantial increase in IPA post-chemotherapy also showed a sustained reduction in monocyte counts at T2 (**Figure 5 for reviewers. 5A-B**). To emphasize this subgroup, we have highlighted these patients in red in both IPA and monocytes changes figures (**Figure 5 for reviewers. 5A-B**). Accordingly, we have revised the manuscript to include this information on **page 18, lines 483-484**: “Patients with substantial increase in IPA level at T2 showed a corresponding reduction of monocyte at T2 (**Supplementary Figure. 8F-G**)”.

Figure 5 for reviewers. IPA increases post chemotherapy in subset of CRC patients. (A) IPA is significantly increased post- chemotherapy in CRC patients. (B) Patients with the strongest increase in IPA show monocyte decrease. (A) and (B) Highlighted in red are the patients with the IPA increase and monocytes reduction. Data are analyzed using Wilcoxon rank-sum test (A) and two-tailed paired T-test (B).

Reviewer #2 (Remarks to the Author):

The manuscript demonstrates that standard CRC chemotherapy with 5-fluorouracil and oxaliplatin induces intestinal mucositis, which alters the gut microbiota composition, leading to increased levels of the bacterial metabolite indole-3-propionic acid (IPA). Elevated IPA reprograms bone marrow myelopoiesis, reducing immunosuppressive Ly6Chigh monocytes and promoting anti-tumor CD4+ T cell responses. This effect, termed "chemomemory" by the authors, reduces colorectal cancer liver metastasis in mice even after chemotherapy has cleared. IPA supplementation alone can mimic this protective effect, suggesting that it could be used to enhance the efficacy of chemotherapy. Analysis of 14 CRC patients revealed that chemotherapy increases serum IPA levels in some individuals, with IPA inversely correlating with circulating monocytes, supporting its clinical relevance. The manuscript contains important data on a timely aspect of CRC and metastases. I would encourage the authors to clarify some key points and temper rather speculative interpretations before publication.

We thank the reviewer for their positive comments that help improve the manuscript and its clarity.

Major

Results

In the section "Anti-metastatic chemomemory is microbiota-dependent": Is mucositis itself microbiota-dependent in their model? Can the authors assess this from their metastasis experiments? While this may not be central to the paper's claims, it could help clarify the proposed mechanism.

We agree that, although not central to the conclusions of our study, this question is of both interest and importance. Chemotherapy-induced intestinal mucositis is partially microbiota-dependent. Highly proliferative intestinal epithelial cells in the crypts are the direct targets of FO chemotherapy, leading to initial epithelial damage. In addition, microbiota amplifies chemotherapy-induced mucositis (7–10). For example, co-administration of antibiotics with 5-FU or doxorubicin reduced intestinal inflammation and immune cell infiltration, leading to less severe mucositis, although microbiota depletion did not prevent direct crypt cell death (11). In addition, we show that FO treatment actively alters the microbiota composition, promoting both the expansion of tryptophan-metabolizing bacteria and Proteobacteria (**please refer to Figure 2F in the manuscript**), the known drivers mucosal inflammation (12).

While our metastatic model was not designed to directly analyze microbiota dependence of mucositis during the acute post-chemotherapy phase—since this would preclude assessing long-term metastatic outcomes—we agree that microbiota-driven mucosal inflammation likely contributes to the changes observed, for example by delaying the recovery of enterocyte absorptive capacity. We now acknowledge this possibility in the limitations section (**page 22, lines 595-597 in the revised manuscript**): “The extent to which intestinal mucositis depends on the presence or the initial composition of the microbiota was not assessed in the metastatic model and warrants further investigation”.

As the central novel claim of our paper is the normalizing effect of IPA of pathological myelopoiesis, we concentrated our effort in revision on understanding the IPA effects in bone marrow. Based on additional experiments we now show that:

1. IPA treatment significantly increases the frequency of macrophages in the bone marrow, while the total abundance of neutrophils and overall monocytes remains unchanged (**Figure 6 for reviewers. 6A**).
2. IPA significantly increases the proportion of quiescent common myeloid progenitors (CMPs) compared to other progenitor populations (**Figure 6 for reviewers. 6B**).
3. CMPs isolated from IPA-treated mice with liver metastases generate more macrophages and fewer CCR2⁺ monocytes in the bone marrow, resulting in decreased circulating monocytes (**Figure 6 for reviewers. 6D-E**).

Figure 6 for reviewers. IPA rewires CMP to mature into macrophages in the bone marrow and reduces monocyte release in the circulation. **(A)** IPA increases the abundance of macrophages in the bone marrow of metastasis bearing mice. $n = 4$ per groups. **(B)** CMP (Lineage⁻cKit⁺Sca1⁻(KLS⁻)CD16/32^{low}CD34⁺) in IPA treated mice are more quiescent. Control, $n = 5$. IPA, $n = 5$. GMP (KLS⁻CD16/32^{low}CD34⁺). **(C)** Scheme of the experiment to analyze the fate of CMP treated with IPA in vivo. **(D)** CMP from IPA treated mice produce more macrophages in the bone marrow of recipient mice. Data show the percentage among CD45.1+ cells. Control, $n = 5$. IPA, $n = 4$. **(E)** CMP from IPA treated mice release less monocytes in the blood. Experiment 1: Control, $n = 5$. IPA, $n = 4$; Experiment 2: Control, $n = 3$. IPA, $n = 4$. Data are shown as mean \pm SD and analyzed using two-tailed unpaired Student's t-test **(A, D, E)** and two-way Anova with Bonferroni's multiple comparisons test **(B)**.

“We tested the effects of high physiological (10 μ M) and supraphysiological (50, 100 and 200 μ M) IPA concentrations (43, 44). Only non-physiological concentration of 100 μ M IPA inhibited growth of patient-derived MSS tumoroids” – How are “physiological” and “supraphysiological” concentrations defined? What evidence supports 10 μ M as physiological in the tumor microenvironment?

Thank you for raising this important question. Physiological IPA levels have been assessed in humans and in animal models previously, and we found them consistent with our own results.

The concentration of IPA in patients' blood ranges from 1 to 10 μM (13–15), while in mice in vivo concentrations of IPA of 7-30 μM are functionally important for nerve regeneration, and between 3 and 6 μM in a sepsis model (16, 17). In our own targeted analysis of healthy volunteers, blood concentrations of IPA ranged from 2 to 7 μM and 1 to 4 μM in control mice, while in mice treated with FO, levels ranged from 6 to 20 μM .

To clarify this point, we changed the text, included the references and explained further the choice of concentration used in the text (**page 10, lines 258-261 in the manuscript**).

“IPA was unable to suppress metastatic growth or tumor cell proliferation in NSG mice (Fig. 4L-O). These results show that in vivo IPA anti-metastatic activity is primarily driven through the immune system.” – I recommend more cautious wording here, as NSG mice differ from immunocompetent models on multiple levels.

We agree that NSG mice differ from immune competent models in many aspects. Therefore, we took the advice the reviewer and changed the statement as follows: “These results indicated IPA anti-metastatic activity is in part mediated by its effect on immune cells”. Please refer to **page 10, lines 268-269** in the manuscript.

Related to Figure 7A-E: Did IPA treatment have any effect on overall tumor size? If not, can the authors speculate why, and could they show the data?

We performed additional analysis on chemotherapy experiment as requested. The overall tumor size was significantly reduced by IPA, chemotherapy and combined treatment (**Figure 7 for reviewers. 7A**). The individual metastasis size was significantly reduced by IPA and chemotherapy, while the difference between chemotherapy and combined treatment was not statistically significant when comparing across the 4 conditions (**Figure 7 for reviewers. 7B-C**). However, pairwise comparison of FO to FO+IPA shows that addition of IPA leads to 50% reduction in the individual metastasis size (**Figure 7 for reviewers. 7B-C**).

We have incorporated the results in **Fig. 7D-E** and added text the following text: “Furthermore, combinatorial FO+IPA therapy significantly decreased metastatic burden and increased necrotic areas within a proportion of liver metastatic lesions (**Fig. 7D-G**)”. This can be found on **page 17, lines 446-448** in the manuscript.

Figure 7 for reviewers. (A-B) FO+IPA reduce liver metastasis load and metastatic size. (C) FO+IPA decrease metastasis size by 50% compared to FO alone. Data are shown as mean \pm SD and analyzed using one-way Anova with Tukey's multiple comparisons test (A, B) two-tailed unpaired Student's t-test (C).

“CRC patients present with lower IPA level as compared to the healthy population, suggesting a role for IPA in disease control and that IPA-producing bacteria are preferentially targeted by antibiotics used to treat these patients.” This interpretation appears highly speculative. Even if the observation holds in larger cohorts, reduced IPA levels could have multiple causes. I suggest a more careful phrasing.

We agree with the reviewer that this claim is speculative, and many confounders could explain this variability. Therefore, we took the advice of the reviewer, and we rephrased this statement as follows: “Although further studies are necessary, we speculate that lower IPA levels in CRC patients may reflect depletion of IPA-producing bacteria by commonly used antibiotics.” Please refer to **page 18, lines 472-474** in the manuscript.

Discussion

5FU/Oxaliplatin is the standard adjuvant therapy for stage III CRC, yet many patients still develop metastases. Can the authors reflect on why the mechanisms they uncover might fail in a significant proportion of patients?

Thank you for this thoughtful suggestion. Our findings indicate that chemotherapy can induce a metastasis-refractory state through microbiota-dependent production of IPA, which alters myelopoiesis and enhances anti-tumor immunity. However, we observed considerable variability in post-chemotherapy IPA levels among patients, suggesting that this mechanism may be compromised in a subset of individuals. This could be due to inter-individual differences in microbiota composition and function, resulting in insufficient IPA production.

Additional extrinsic factors such as diet and medications may further hinder microbiota recovery or microbial metabolism (18, 19). For these reasons, it is important to explore novel strategies that leverage microbial products—such as postbiotics and other microbiota-based therapies containing IPA—to enhance and potentiate existing treatments.

We have now included the following statement in the manuscript (**page 20, lines 528-532**): “The high variability of IPA levels observed in patients suggests the presence of other patient-specific factors that may limit this protective effect. These include differences in microbiota composition and function, diet, and concomitant medications, all of which can result in insufficient IPA production. Understanding these variables could inform the development of more personalized adjuvant strategies based on microbial products.”

Methods

It appears from the main Methods section that only GraphPad was used for statistical analyses. Is this sufficient for the more complex data, such as proteomics and spatial analyses?

We thank the reviewer for highlighting this point. The statistical methods for each complex data analysis, such as RNA-seq, metabolomics, and spatial analysis, were detailed in their respective sections. To improve clarity, we have now added additional information in the general statistics section of the Methods, referencing that the specific details can be found within each individual section. We have included the following information in the statistics part of methods: “The statistical methods for each complex data analysis, such as RNA-seq, metabolomics, and spatial analysis, are detailed in their respective sections.” This can be found on **page 36, lines 966-968**.

Minor

Introduction

Consider rewording: stage I & II CRC are still “invasive”; consider removing “Non-invasive.” “antibiotics prior or during chemotherapy treatment significantly reduced disease-free survival in CRC patients (7), suggesting that gut microbiota confers protection against metastatic recurrence.” – Given the potential for many confounding factors, I suggest more cautious phrasing. For example:

We thank the reviewer with this suggestion and changed the text accordingly on **page 3, line 61 and on page 3, lines 80-81**:

“...in CRC patients. However, the mechanistic link between chemotherapy, microbiota alterations and cancer progression, including liver metastasis development, remains incompletely understood.”

“indicating that the metastasis refractory state is long-lasting (Fig. 1L-N; Fig. S1JK). In all, we demonstrate that FO induced mucositis and systemic effect maintain a lasting anti-tumor effect.” The authors use “long-lasting” for a period of ten days. Would “lasting” suffice?

We agree with the reviewer. We replaced long lasting with lasting in the revised version of our manuscript.

“...suppressed liver metastasis growth to levels similar to those observed in FO-treated, microbiota-competent mice (Fig. 4IJ; Fig. 1H-I).” – As the authors did not measure growth dynamically, it would be more accurate to use “liver metastasis size” or “burden.”

We have modified the text as proposed by the reviewer replacing “growth” with “burden”.

The Supplementary Methods should be clearly referenced, or, as is more typical for the journal, all methods should be included in the main text.

We have changed the method part accordingly.

Reviewer #3 (Remarks to the Author):

The manuscript by Bersier et al. proposes a mechanism how chemotherapy effect systemic immune memory by regulating intestinal microbiota and microbiota-dependent metabolite IPA, which in turn modulates myelopoiesis and subsequent T cell activation suppressing liver metastasis development. While the role of IPA in regulation of T cell immune responses in anti-cancer treatment was previously suggested, this work provides novel insights into the indirect, monocyte-dependent control of anti-tumor T cell responses in liver metastases. The manuscript provides strong experimental evidence supporting proposed mechanisms and use various loss- of- function and gain -of-function approaches and multiple metastasis models to support conclusions.

Overall, this is an exciting study. However, some additional clarifications can further strengthen presented work.

We thank the reviewer for positive and enthusiastic evaluation of our manuscript.

1. What would be the explanation on why antibiotic treatment does not suppress metastasis development (Fig 2)?

We thank the reviewer for this comment. We do not expect antibiotics treatment to suppress metastasis in our model. Indeed, we found that chemotherapy increases circulating levels of IPA derived from gut microbiota, and IPA reduces accumulation of immunosuppressive myeloid cells, this is the reason why antibiotics reduced the chemomemory but are not directly affecting tumor burden of mice receiving control treatment. To clarify this point, we provide now a summary model of the main conclusion of our work **on page 16, lines 433-438** and the summary model is in **Figure 8 for reviewers** and **Fig. 6M** in the revised manuscript.

Figure 8 for reviewers. In the absence of IPA, tumor induced $Ly6c^{high}CCR2+$ monocytes are released from bone marrow in the circulation and recruited to the tumor microenvironment, where they suppress antitumor immunity. IPA promotes maturation of these monocytes, decreasing their release in circulation. This in turn relieves immunosuppression and enhances antitumor immunity mediated by $CD4+Th1$ cells in metastatic niche, driving IPA-mediated chemomemory.

2. Metastasis models are relatively short. Is this effect of chemotherapy on microbiota and subsequent immune system rewiring long lasting? It would be important to discuss it in the manuscript.

We thank the reviewer for this suggestion. In the manuscript, we show the chemomemory effect 5 and 10 days after chemotherapy. We now analyzed 20 days post chemotherapy time point, where we find a tendency towards limitation of metastasis, suggesting a wearing of the effect of the one cycle of chemo between 10 and 20 days (**Figure 9 for reviewers. 9A-B**). Interestingly, a typical chemotherapy cycle lasts two weeks, and patients usually undergo multiple cycles. It will be important to determine in future studies whether repeated cycles lead to sustained high levels of IPA, long-lasting immune system rewiring, and potentially even stronger control of metastasis. We have included these new data (**Supplementary Fig. 2C-E**) and the following description of the result **on page 6, lines 144-147**: “Injecting MC-38-GFP tumor cells twenty days after FO pre-conditioning, mimicking the end of a treatment cycle, showed a tendency for persistence of metastasis refractory state (**Supplementary Fig. 2C-E**).” Additionally, the corresponding discussion can be found in the revised manuscript **on page 19, lines 514-518** as follows: “A typical chemotherapy cycle lasts two weeks, and patients usually undergo 6 to 12 cycles of FOLFOX treatment. We found that the metastasis-refractory state lasts up to 10–20 days after a single FO cycle (**Fig. 1L-N; Supplementary Fig. 2B-E**). It will be important to determine in future studies whether repeated cycles lead to sustained high levels of IPA, prolonged immune system rewiring, and potentially stronger control of metastasis.”

Figure 9 for reviewers. FO chemomemory is maintained in some mice 20 days post FO treatment. **(A)** Quantification of percentage and **(B)** count of MC38-GFP cancer cells in liver of mice in the indicated conditions. Control, n = 4. FO, n = 5. Data are shown as mean ± SD and analyzed using two-tailed unpaired Student’s t-test (**A, B**).

3. Can authors clarified how would depletion of circulating monocytes affect chemotherapy /IPA mediated T cell immune responses in metastases?

Thank you for this comment. Our model is that chemotherapy expands tryptophan-metabolizing bacteria, leading to elevated IPA levels. In the bone marrow, IPA drives myeloid

cell maturation toward macrophages and reduces the production and release of Ly6c^{high} monocytes, thereby limiting the infiltration of immunosuppressive myeloid cells into the tumor microenvironment. Additionally, this leads to increase proportion of myeloid cells expressing MHCII in the liver of IPA treated mice (please refer to **Supplementary Fig. 6D**). These changes enhance CD4⁺ T cell activity, ultimately limiting tumor progression in the liver. The corresponding model is now included on **page 16, lines 433-438** of the revised manuscript (**Fig. 6M**).

To further study the reason for reduction in circulating Ly6c^{high} monocytes, we performed a CMP adoptive transfer experiment. We induced metastases in CD45.1 mice and treated them with either control or IPA daily treatment (**Figure 10 for reviewers. 10C**). We then FACS-sorted CMPs and adoptively transferred them into CD45.2 recipient mice, tracking CMP fate at early time points in vivo using established protocols (1, 2). The proportion of CD45.1-derived cells in the bone marrow was similar between groups, and we did not detect significant differences in the frequencies of monocytes or neutrophils derived from transferred CMPs. In contrast, CMPs from IPA-treated mice generated a higher proportion of macrophages in the bone marrow and fewer CCR2-expressing monocyte (**Figure 10 for reviewers. 10D**). Several studies have shown that blocking monocyte migration—either through CCR2 inhibition or using CCR2 knockout mouse models—enhances T cell-mediated immunity (20, 21). For example, in a model where myocardial infarction promoted the expansion of immunosuppressive Ly6c^{high} monocytes and tumor progression, depletion of these cells via a CCR2 inhibitor helped control tumor growth by enhancing CD8⁺ T cell-mediated immunity (22). Consistent with these data, we observed fewer monocytes in the blood of mice transferred with CMPs from IPA treated donors (**Figure 10 for reviewers. 10E**). Altogether, these results indicate that CMPs are the main target of IPA in bone marrow and that IPA promotes differentiation of CMPs towards macrophage lineage, while reducing release of monocytes in the periphery. **These new data are included in the revised manuscript on page 16, lines 414-438 (in Fig. 6 and Supplementary Fig. 7)**, and we propose a model in which IPA promotes pre-mature CMP differentiation and retention in bone marrow. In turn, the reduction in circulating monocytes leads to decreased monocyte infiltration into the TME and improved anti-tumor immune response (**Fig. 6M**).

Figure 10 for reviewers. IPAs rewires CMP to mature into macrophages in the bone marrow and reduces monocyte release in the circulation. **(A)** IPAs increase the abundance of macrophages in the bone marrow of metastasis-bearing mice. $n = 4$ per groups. **(B)** CMP (Lineage⁻cKit⁺Sca1⁻(KLS⁻)CD16/32^{low}CD34⁺) in IPAs-treated mice are more quiescent. Control, $n = 5$. IPAs, $n = 5$. GMP (KLS⁻CD16/32^{low}CD34⁺). **(C)** Scheme of the experiment to analyze the fate of CMP treated with IPAs in vivo. **(D)** CMP from IPAs-treated mice produce more macrophages in the bone marrow of recipient mice. Data show the percentage among CD45.1⁺ cells. Control, $n = 5$. IPAs, $n = 4$. **(E)** CMP from IPAs-treated mice release less monocytes in the blood. Experiment 1: Control, $n = 5$. IPAs, $n = 4$; Experiment 2: Control, $n = 3$. IPAs, $n = 4$. Data are shown as mean \pm SD and analyzed using two-tailed unpaired Student's t -test (**A**, **D**, **E**) and two-way Anova with Bonferroni's multiple comparisons test (**B**).

4. Fig 6 E. The gates defining neutrophils need to be corrected. Typically neutrophils are defined as Ly6G^{hi}, Ly6C⁺, unlike presented on this figure.

We thank the reviewer for this comment and provide now this figure. These data are found now in **Supplementary Fig. 7D-E** in the revised manuscript and presented here in **Figure 11 for reviewers**.

Figure 11 for reviewers. IPA does not affect neutrophils while Ly6c^{high} monocytes are reduced. **(A)** Percentage of neutrophils generated in the presence of vehicle (DMSO) or IPA in vitro. n= 8 per condition. Data show biological replicate pooled from two independent experiments **(B)** Gating strategy. Data are shown as mean ± SD and analyzed using repeated measure one-way Anova with Tukey's multiple comparisons test **(A)**.

Reviewer #4 (Remarks to the Author):

Bersier et al. analyzed how chemotherapy-induced normal tissue injury can influence individual susceptibility to metastasis, specifically in colorectal cancer (CRC). The authors showed that chemotherapy indirectly prevents liver metastases by inducing a long-lasting systemic "chemomemory." This effect was mediated by intestinal mucositis, which altered gut nutrient availability and promoted the expansion of tryptophan-metabolizing bacteria, leading to increased production of the microbial metabolite indole-3-propionic acid (IPA). They further showed that IPA reshaped bone marrow myelopoiesis, resulting in reduced numbers of immunosuppressive Ly6C-high monocytes in the circulation and liver. This reduction facilitated the accumulation of CD4⁺ Th1-like T cells and enhanced colocalization of CD8⁺ cytotoxic T cells, ultimately strengthening anti-tumor immunity

We thank the reviewer for careful revision of our work and valuable comments.

General Remark 1: The authors have went to great lengths to describe the beneficial role of IPA in reinforcing the chemomemory in the context of 5-FU and Oxali treated CRC-bearing animals. The results are in this regard convincing, however they are not novel as this has been reported by other scientists recently (https://ascopubs.org/doi/10.1200/JCO.2024.42.16_suppl.e15194; PMID: 38490195).

We thank the reviewer for this comment. Indeed, the ASCO abstract highlights the importance of IPA in enhancing 5-FU efficacy in colorectal cancer. However, since this study has not yet undergone peer review or been published, we are unable to fully evaluate their data. Nonetheless, we find the primary observation plausible and exciting, as it aligns with our study supporting a role for IPA in modulating chemotherapy response.

That said, based on the information provided in the abstract, we did not observe a direct effect of IPA on CD8⁺ T cells in our experiments (**Figure 12 for reviewers. 12H-P**). Specifically, treatment with IPA at physiological concentration did not influence CD8⁺ T cell proliferation or activation, as assessed by cytokine production.

Figure 12 for reviewers. From supplementary Figure 5. (H) Scheme of the experiment to analyze the effect of IPA on CD4+ and CD8+ T cells. **(I)** IPA does not directly affect proliferation of activated CD4+ and CD8+ T cells in vitro. Histogram of CFSE dilution in CD4+ (left) and CD8+ (right) cells. **(J)** Quantification of CD4+ and CD8+ proliferated cells in the indicated conditions. n = 3 per condition. **(K)** Physiological concentration of IPA does not influence CD4+ and CD8+ T cell activation in vitro. Concentration of IFN γ and TNF α in cell culture supernatant collected three days after exposure. n= 3 to 4 biological replicates per condition. **(L)** Representative image of CD8+TCF1+ infiltration in metastatic lesions. Staining for CD8 (green), TCF1 (magenta) and DNA (blue). Scale bar: 50 μ m. **(M)** IPA does not change peritumoral CD8+TCF1+ T cell infiltration. Quantification of CD8+TCF1+ cells per metastatic area. n = 4 per condition. **(N)** IPA does not increase CD8+ exhausted progenitor cells. Histogram of TCF1 expression in CD8+PD1+ T cells. **(O)** Quantification of the percentage of CD8+PD1+TCF1+ cells. **(P)** Quantification of TCF1 MFI in CD8+PD1+ T cells. n= 4 biological replicates per conditions. Data are shown as mean \pm SD and analyzed using two-tailed unpaired Student's t-test **(M)** and one-way Anova with Tukey's multiple comparisons test **(J, K, O, P)**.

We have discussed PMID: 38490195 in both the Results (on pages 10-11, lines 270-281 in the revised manuscript) and Discussion sections (on page 20, lines 533-540). Importantly, in our hands, IPA did not directly increase TCF1 expression in CD8+ T cells, either in vitro at physiological or supraphysiological concentrations (**Figure 12 for reviewers. 12H-P**) or in vivo within our metastatic model. Instead, we demonstrate that myeloid progenitor cells, rather than T cells, are the primary cellular targets of IPA in cancer. We performed additional experiments and now show that:

4. IPA treatment significantly increases the frequency of macrophages in the bone marrow, while the total abundance of neutrophils and overall monocytes remains unchanged (**Figure 13 for reviewers. 13A**).

- IPA significantly increases the proportion of quiescent common myeloid progenitors (CMPs) compared to other progenitor populations (**Figure 13 for reviewers. 13B**).
- CMPs isolated from IPA-treated mice with liver metastases generate more macrophages and fewer CCR2⁺ monocytes in the bone marrow, resulting in decreased circulating monocytes (**Figure 13 for reviewers. 13D-E**).

Figure 13 for reviewers. IPA rewires CMP to mature into macrophages in the bone marrow and reduces monocyte release in the circulation. **(A)** IPA increases the abundance of macrophages in the bone marrow of metastasis bearing mice. $n = 4$ per groups. **(B)** CMP (Lineage⁻cKit⁺Sca1⁻(KLS⁻)CD16/32^{low}CD34⁺) in IPA treated mice are more quiescent. Control, $n = 5$. IPA, $n = 5$. GMP (KLS⁻CD16/32^{low}CD34⁺). **(C)** Scheme of the experiment to analyze the fate of CMP treated with IPA in vivo. **(D)** CMP from IPA treated mice produce more macrophages in the bone marrow of recipient mice. Data show the percentage among CD45.1⁺ cells. Control, $n = 5$. IPA, $n = 4$. **(E)** CMP from IPA treated mice release less monocytes in the blood. Experiment 1: Control, $n = 5$. IPA, $n = 4$; Experiment 2: Control, $n = 3$. IPA, $n = 4$. Data are shown as mean \pm SD and analyzed using two-tailed unpaired Student's t-test (**A, D, E**) and two-way Anova with Bonferroni's multiple comparisons test (**B**).

These new results refine our initial model and indicate that CMPs are the main target of IPA in the bone marrow. IPA promotes CMP differentiation toward the macrophage lineage

while reducing the release of monocytes into the periphery. As CCR2 has been previously shown to be essential for monocyte egress from the bone marrow (20, 21), the observed reduction in CCR2+ monocytes likely contributes to this effect. These new data are included in the revised manuscript on **pages 16, lines 414-438 (Fig. 6; Supplementary Fig. 7)**. We consider them both novel and important, as they identify a previously unrecognized cellular target of IPA that will help guiding future research.

General Remark 2: Despite very large animal experiment, the mechanism on how IPA actually works in the cell (eventual receptor or intracellular target) remain unknown.

We thank the reviewer for this valuable comment. We agree that elucidating the molecular mechanism underlying IPA molecular effectors in myeloid progenitors and myeloid cells is an important next step, especially for therapeutic applications. While many studies have linked IPA to AHR signaling, our experiments show that IPA does not alter AHR activity in bone marrow cells at 10 μ M, as demonstrated by the lack of induction of AHR target gene signatures (**Figure 14 for reviewers**).

Figure 14 for reviewers. AHR signature is not increased in bone marrow cells treated by IPA.

These results are in line with previous biochemical analyses demonstrating that IPA is unable to activate AHR in AHR reporter assay *in vitro* or compete with a *bona fide* AHR ligand in direct binding assay (**Figure 15 for reviewers. 15A-B**) (15).

Figure 15 for reviewers. Taken from(15). **(A)** IPA does not activate AHR in stably transfected luciferase reporter construct Hepa1.1 cell line. **(B)** In human CRC cells, IPA physiological concentration does not induce AHR gene transcription.

We have begun identifying potential IPA interactors using in silico modelling and chemoproteomic approaches. Fully characterizing the as-yet unknown intracellular targets and receptors of a small molecule such as IPA will likely require several years and falls beyond the scope of the current manuscript. Nevertheless, we include negative data demonstrating the absence of an AHR signature in IPA-treated cells and discuss IPA's established inability to activate AHR (15) on page 15, lines 396-399 (Supplementary Fig. 7F in the revised manuscript). We believe these data are important, as they help narrow the search for relevant intracellular targets and avoid misattribution of IPA's effects to the AHR pathway in bone marrow cells. We also include the reviewer's comment in the limitation section on page 22, lines 594-597:

« The molecular targets and effectors of IPA in myeloid progenitors, as well as its effects on other hematopoietic and metastatic niche components, remain to be identified. »

General Remark 3: The metabolomics analysis has not been well detailed, targeted approach needs the specific information on which transitions have been used, retention times, etc. More detail is needed if these results are to be reproduced in the future.

Thank you for the comments, we now included more detailed information for targeted approach as follows (on pages 27-28, lines 727-740 of the revised manuscript):

“The absolute serum concentrations of 3-indolelactic acid (m/z 204→142, RT=2.6), 3-indolepropionic acid (m/z 188→59, RT=3.5), and tryptophan (m/z 205→188, RT=0.93) were quantitated at Metabolon facility by LC-MS. Serum samples were spiked with stable labeled internal standards, homogenized, and subjected to protein precipitation with an organic solvent. The sample is centrifuged, and a portion of the supernatant was injected onto an Agilent 1290/Sciex QTrap 5500 LC-MS/MS system equipped with a C18 reversed phase ultrahigh performance liquid chromatography column. The mass spectrometer was operated in positive mode for tryptophan and negative mode for the remaining analytes using electrospray ionization (ESI). The peak area of the individual analyte product ions was measured against the peak area of the product ions of the corresponding internal standards. Quantitation was performed using a weighted linear least squares regression generated from fortified calibration standards prepared immediately prior to each run. The LC-MS/MS raw data is collected and processed using SCIEX software Analyst 1.6.3 and processed using SCIEX OS-MQ software.”

Reviewer #5 (Remarks to the Author):

We thank the reviewer for participating in the reviewing of our work and for their valuable comments.

References

1. Cortez-Retamozo V, et al. Origins of tumor-associated macrophages and neutrophils. *Proc Natl Acad Sci*. 2012;109(7):2491–2496.
2. Yáñez A, et al. Granulocyte-Monocyte Progenitors and Monocyte-Dendritic Cell Progenitors Independently Produce Functionally Distinct Monocytes. *Immunity*. 2017;47(5):890-902.e4.
3. Bailly-Botuha C, et al. Plasma Citrulline Concentration Reflects Enterocyte Mass in Children With Short Bowel Syndrome. *Pediatr Res*. 2009;65(5):559–563.
4. Crenn P, Messing B, Cynober L. Citrulline as a biomarker of intestinal failure due to enterocyte mass reduction. *Clin Nutr*. 2008;27(3):328–339.
5. Wang B, et al. Metabolism pathways of arachidonic acids: mechanisms and potential therapeutic targets. *Signal Transduct Target Ther*. 2021;6(1):94.
6. Kalinski P. Regulation of Immune Responses by Prostaglandin E2. *J Immunol*. 2012;188(1):21–28.
7. Anderson CJ, et al. Metabolite-based inter-kingdom communication controls intestinal tissue recovery following chemotherapeutic injury. *Cell Host Microbe*. 2024;32(9):1469-1487.e9.
8. Hamouda N, et al. Apoptosis, Dysbiosis and Expression of Inflammatory Cytokines are Sequential Events in the Development of 5-Fluorouracil-Induced Intestinal Mucositis in Mice. *Basic Amp Clin Pharmacol Amp Toxicol*. 2017;121(3):159–168.
9. Li H-L, et al. Alteration of Gut Microbiota and Inflammatory Cytokine/Chemokine Profiles in 5-Fluorouracil Induced Intestinal Mucositis. *Front Cell Infect Mi*. 2017;7:455.
10. Pedroso SHSP, et al. Evaluation of mucositis induced by irinotecan after microbial colonization in germ-free mice. *Microbiology*. 2015;161(10):1950–1960.
11. Carr JS, King S, Dekaney CM. Depletion of enteric bacteria diminishes leukocyte infiltration following doxorubicin-induced small intestinal damage in mice. *PLoS ONE*. 2017;12(3):e0173429.
12. Shin N-R, Whon TW, Bae J-W. Proteobacteria: microbial signature of dysbiosis in gut microbiota. *Trends Biotechnol*. 2015;33(9):496–503.
13. Wang Y-C, et al. Indole-3-Propionic Acid Protects Against Heart Failure With Preserved Ejection Fraction. *Circ Res*. 2024;134(4):371–389.
14. Tuomainen M, et al. Associations of serum indolepropionic acid, a gut microbiota metabolite, with type 2 diabetes and low-grade inflammation in high-risk individuals. *Nutr Diabetes*. 2018;8(1):35.
15. Morgan EW, et al. Contribution of Circulating Host and Microbial Tryptophan Metabolites Toward Ah Receptor Activation. *Int J Tryptophan Res*. 2023;16:11786469231182510.
16. Serger E, et al. The gut metabolite indole-3 propionate promotes nerve regeneration and repair. *Nature*. 2022;1–8.

17. Heumel S, et al. Shotgun metagenomics and systemic targeted metabolomics highlight indole-3-propionic acid as a protective gut microbial metabolite against influenza infection. *Gut Microbes*. 2024;16(1):2325067.
18. Gacesa R, et al. Environmental factors shaping the gut microbiome in a Dutch population. *Nature*. 2022;604(7907):732–739.
19. Zhou X, et al. Longitudinal profiling of the microbiome at four body sites reveals core stability and individualized dynamics during health and disease. *Cell Host Microbe*. 2024;32(4):506-526.e9.
20. Sanford DE, et al. Inflammatory Monocyte Mobilization Decreases Patient Survival in Pancreatic Cancer: A Role for Targeting the CCL2/CCR2 Axis. *Clin Cancer Res*. 2013;19(13):3404–3415.
21. Flores-Toro JA, et al. CCR2 inhibition reduces tumor myeloid cells and unmasks a checkpoint inhibitor effect to slow progression of resistant murine gliomas. *Proc Natl Acad Sci*. 2020;117(2):1129–1138.
22. Koelwyn GJ, et al. Myocardial infarction accelerates breast cancer via innate immune reprogramming. *Nat Med*. 2020;26(9):1452–1458.

We thank the Reviewers for their positive feedback and thoughtful evaluation of our manuscript.

Comments are in blue, and our responses are written in black.

POINT BY POINT RESPONSES TO REVIEWERS' COMMENTS

Reviewer #1 (Remarks to the Author):

The authors have responded appropriately to my comments. I have no further comments

We thank Reviewer 1 for their initial comments and are pleased that we have satisfactorily addressed their concerns in the revised version of the manuscript.

Reviewer #2 (Remarks to the Author):

Thank you for your thorough revision of the manuscript. All points raised in my previous review were addressed, and I do not have any further comments.

We thank Reviewer 2 for their initial comments and are pleased that we have satisfactorily addressed their concerns in the revised version of the manuscript.

Reviewer #3 (Remarks to the Author):

Authors successfully addressed previously raised concerns and revised the manuscript accordingly. I have no further questions.

We thank Reviewer 3 for their initial comments and are pleased that we have satisfactorily addressed their concerns in the revised version of the manuscript.

Reviewer #4 (Remarks to the Author):

Bersier et al. have submitted a revised manuscript that addresses some of the reviewers' questions and provides additional clarity on others; however, several issues remain unresolved. The primary concern—the lack of mechanistic insight into the action of IPA in myeloid cells—remains pertinent. This is particularly important given that the present data appear to conflict with previous findings (PMID: 38490195), where IPA targets different cells via a distinct mechanism. While this does not invalidate the authors' findings, it highlights the need for more definitive mechanistic understanding, especially in light of multiple reports demonstrating the beneficial role of IPA in cancer therapy.

We thank reviewer 4 for their comments. As stated in our previous rebutal, we agree that elucidating the molecular mechanism underlying IPA molecular effectors in myeloid progenitors and myeloid cells is an important next step, especially for therapeutic applications. However, fully characterizing the as-yet unknown intracellular targets and receptors of a small molecule such as IPA will likely require several years and falls beyond the scope of the current manuscript.

A second major concern pertains to the metabolomics analyses presented throughout the manuscript, including key figures (Figs. 2, 3, and 8), particularly those regarding IPA concentrations in patients. Based on the additional details provided, it is evident that the metabolomics analyses were conducted following the methodology of Fuhrer et al., 2011 (as per authors information), which no longer reflects current standards. Today, it is considered

unacceptable to quantify or identify metabolites—as in Figure 3F (untargeted)—without the use of liquid chromatography (retention times are unavailable) and MS² validation. Direct injection and MS¹-based quantification may be acceptable in limited in vitro screening experiments with defined metabolite pools but not in complex biological matrices such as serum or cellular extracts.

We respectfully disagree with the Reviewer's opinion that "Today, it is considered unacceptable to quantify or identify" metabolites in absence of chromatography. In untargeted metabolomics, quantification is based on MS¹ scans – and FIA is by no mean inferior to (untargeted) LC-MS. The CVs are frequently better than with LC-MS (typically <10% across the dynamic range) because they don't suffer from the errors associated with integrating 2D-chromatograms, that in metabolomics often feature asymmetric shapes and tails. Yes, FIA doesn't allow resolving isomers, but this is a quest that can be addressed ad-hoc using the proper approach and the proper samples, which is what we did in the case for 3-IPA.

Specifically, we indeed analyzed serum samples using an mixed-mode LC (<https://chemrxiv.org/engage/chemrxiv/article-details/6824924be561f77ed4c75578>) using an Agilent 6546 QTOF. At the expected m/z of 188.0717, we observed only a single peak at low intensity in negative mode, and nothing in positive mode. Because of the low abundance, we struggled to collect an informative MS² spectrum for spectral matching.

Therefore, we moved on to confirm the identity by retention time. According to HMDB, there are 6 isomers with formula C₁₁H₁₁NO₂: indole-3-propionic acid, 2-(1-methyl-1H-indol-3-yl)acetic acid, two indole variants with esterified tails, and two pyrroline derivatives with virtually no anionic group. Because of the preference in ionization in negative mode (over positive), we suspected that the peak could only be indole-3-propionic acid or

2-(1-methyl-1H-indol-3-yl)acetic acid, as all of the remaining candidates didn't have an ionizable carboxylic group. Based on these notions, we opted to use the aforementioned mixed-mode LC system because the stationary phase acts a weak anion exchanger that retains acids, but not amines (see paper). When injecting pure standards of the two candidates, we observed that indole-3-propionic acid had exactly the same retention time observed in the serum samples (< 1 sec deviation), while the other candidate eluted later and was separated at baseline. Note that all other isomers without carboxylic group elute much earlier. As the only HMDB candidate that elutes at the observed time is indole-3-acetic acid, we could confirm its identity regardless of MS² fragmentation.

The identification of IPA was further consolidated by the targeted analyses performed in the

following by Metabolon, using a different method that also matched RT and MRM and, additionally, by the University of Lausanne metabolomics facility.

Most importantly, the role and therefore the identity of IPA was validated in the functional tests that are included in the paper and demonstrate the effect IPA in potentiating the chemotherapeutic effect. As the functional validation is way more compelling than any spectral or analytical evidence, we decided to omit this long explanation from the manuscript, and included the data that initially led us to “discover” IPA, that is FIA-MS.

We have now included the following (page 8, lines 215-226):

“Retention time comparison with IPA pure standard confirmed the IPA identity detected by untargeted FIA (**Supplementary Fig. 4a**).

Analysis of functional microbiome-encoded pathways using PICRUSt2 revealed an enrichment in tryptophan metabolism among taxa increased after FO treatment (**Fig. 3d**)³². Importantly, IPA is a by-product of tryptophan metabolism and produced exclusively by Clostridia and in particular *Clostridiaceae*s, a genus increased upon FO treatment (**Supplementary Fig. 4b**)³³. The initial targeted metabolomics analysis showed an increase of IPA concentration in FO treated serum over the upper detection limit (**Supplementary Fig. 4c**). Consequently, we performed the targeted LC-MS/MS in a second facility for the samples reaching detection limit and this second analysis further validated a >10-fold IPA increase in the majority of FO treated mice (**Fig. 3e**).”

Similarly, the targeted quantification presented in Figure 3E is problematic. Using a single transition, as now described in the Materials and Methods section, is highly unconventional and does not meet current standards for reliable targeted metabolomics.

We are not aware of what Standards the Reviewer refers to. Outside of diagnostic and regulatory fields, it is common to have only one MRM per compound, see <https://pmc.ncbi.nlm.nih.gov/articles/PMC3334318/>, <https://doi.org/10.1016/j.aca.2023.341791>, <https://www.nature.com/articles/nprot.2012.024>, <https://www.sciencedirect.com/science/article/pii/S0021967323005678>, <https://www.mdpi.com/2218-1989/14/11/622#app1-metabolites-14-00622> just to mention some work from various experts/labs in metabolomics. The use of a qualifier MRM is seen only when there is uncertainty in the RT, i.e. like in <https://star-protocols.cell.com/protocols/1665>.

The analysis was done by Metabolon, the leading CRO in the field, based on their internal library of known RT and MRM that are likely unique across their whole database (see more detailed information below). There is no reason to doubt the selectivity of their assays (for compounds with m/z 205, 204, and 188), in particular if we consider this measurement in the context of the whole study, including the earlier LC-MS analysis we did in our lab, validation by another metabolomic facility and the functional validation with IPA.

Finally, it is unclear why the authors have only partially addressed reviewers' requests for detailed MS methodology—a critical component of the study. The description provided (“...was injected onto an Agilent 1290/Sciex QTrap 5500 LC-MS/MS system equipped with a C18 reversed-phase ... column”) lacks essential details regarding the column type, dimensions, part number, gradient conditions, and solvent composition. Reporting retention times without providing these key parameters limits reproducibility and transparency. In conclusion, the metabolomics analyses must be performed according to current state-of-the-art standards. Complete and detailed reporting in the Materials and Methods section is essential to ensure reproducibility and adherence to international guidelines.

We are happy to include in the revised manuscript the following additional information and in Supplementary Data 6 (pages 27-29, lines 711-774):

The absolute serum concentrations of 3-indolelactic acid(m/z 204→142, $RT=2.6$), 3-indolepropionic acid (m/z 188→59, $RT=3.5$), and tryptophan (m/z 205→188, $RT=0.93$) were quantitated at Metabolon facility by LC-MS/MS. Stock and internal standard solutions were prepared by dissolving an appropriate amount of material in ACN: water (1:1). These stock solutions were diluted to the Standard H level and further diluted to each of the other concentration levels. For sample preparation, serum was thawed on ice and vortexed for 10–15 s before aliquoting. For blank, blank-IS, and calibration standards, 0.0500 mL PBS was pipetted into the appropriate wells of a 2 mL 96-well plate. For matrix QC samples, 0.0500 mL QC serum was added, and for study samples, 0.0500 mL serum was dispensed into designated wells. Calibration standards were prepared by adding 0.0200 mL of the corresponding calibration spiking solution. Internal standard working solution (0.0200 mL) was added to calibration standards, blank-IS, QC samples, and study samples. Blank samples received 0.0400 mL ACN/water (1:1), while blank-IS, QC samples, and study samples received 0.0200 mL ACN/water (1:1). Methanol (0.200 mL) was then added to all wells. Plates were capped, vortexed for 1 min, and centrifuged for 10 min at 4000 rpm. A 0.100 mL aliquot of the resulting supernatant was transferred to a fresh 650 μ L 96-well plate, capped, and submitted for LC-MS/MS analysis. SAMPLE ANALYSIS: A 2.0 μ L aliquot of supernatant was injected onto an Agilent 1290/Sciex QTrap 5500 LC-MS/MS system equipped with a Waters Acquity BEH C18 column (1.7 μ m, 2.1 \times 100 mm). Chromatographic conditions were as follows: Column temperature: 50 $^{\circ}$ C, Autosampler temperature: 4 $^{\circ}$ C, Mobile Phase A1: 0.05% formic acid in water, Mobile Phase B1: 0.05% formic acid in acetonitrile, Flow rate: 0.550 mL/min, Injection volume: 2.0 μ L (adjusted as needed for sensitivity), Needle wash: methanol/water (1:1). The MS/MS parameters were as follow Curtain Gas: 35, Collision Gas: Medium, IonSpray Voltage: -4500 V, Temperature: 500 $^{\circ}$ C, Gas 1: 70, Gas 2: 70, Scan Mode: Multiple Reaction Monitoring (MRM), negative mode. Further information can be found in Supplementary Data 6.

The peak area of the individual analyte product ions was measured against the peak area of the product ions of the corresponding internal standards. Quantitation was performed using a weighted linear least squares regression generated from fortified calibration standards prepared immediately prior to each run. The LC-MS/MS raw data is collected and processed using SCIEX software Analyst 1.6.3 and processed using SCIEX OS-MQ software.

Alternatively, targeted analysis of selected serum samples was performed at the metabolomic facility of the University of Lausanne. Serum (20 μ L) was extracted with ice-cold methanol (180 μ L) spiked with internal standards (IS). After centrifugation (4 $^{\circ}$ C, 21,000 g, 15 min), the supernatants were evaporated to dryness, reconstituted in water (50 μ L), vortexed, and sonicated for 3 min. Each extract was then centrifuged again (4 $^{\circ}$ C, 21,000 g, 15 min) and the supernatant transferred to an LC-MS vial for LC-MS/MS analysis as described below. Calibrators were generated following the same procedure, by the addition of methanol spiked with IS mixture to each calibrator (20 μ L), vortexed and transferred to LC-MS vials for the injection.

Extracts were analyzed by Reversed Phase Liquid Chromatography coupled to tandem mass spectrometry (RPLC-MS/MS) on a 6495 triple quadrupole system (QqQ) interfaced with 1290 UHPLC system (Agilent Technologies), in the analytical conditions modified from van der Velpen et al.⁹⁵. Data were acquired in Dynamic Multiple Reaction Monitoring (dMRM) mode with a total cycle time of 400 ms, applying optimized collision energies for each metabolite, using MassHunter (Agilent technologies, version B.07.00). Chromatographic separation was performed on a Acquity HSS T3 (1.8 μ m 2.1 mm \times 100 mm) column (Waters, Ireland), in positive ionization mode. The mobile phase consisted of A = 5 mM ammonium formate and 0.1% formic acid in H₂O and B = 100% methanol. The gradient was as follows: 0 min 0 %B, 2 min 0 %B, 4 min 5 %B, 5 min 10 %B, 7 min 90 %B, 8 min 0 %B, 11 min 0 %B . The flow rate was 300 μ L/min, the column temperature 20 $^{\circ}$ C, and the

injection volume 2 μ L. ESI source parameters were as follows: dry gas temperature 290 $^{\circ}$ C, dry gas flow 14 L/min, nebulizer pressure 45 psi, sheath gas temperature 350 $^{\circ}$ C, sheath gas flow 12 L/min, nozzle voltage 500 V, and capillary voltage +4000 V. Further information can be found in Supplementary Data 6.

Raw LC-MS/MS data were processed using MassHunter Quantitative Analysis 10.0. Absolute quantification of indole metabolites was performed based on calibration curves and the stable isotope-labeled internal standards (IS) which were used to determine the response factor for each metabolite. Linearity of the standard curves was evaluated for each metabolite using a ten-point range; in addition, peak area integration was manually curated and corrected where necessary.

Reviewer #5 (Remarks to the Author):

We thank Reviewer 5 for their initial comments and are pleased that we have satisfactorily addressed their concerns in the revised version of the manuscript.

Reviewer #6 - Arbitrating Reviewer

So, in summary, in my view the authors have sufficiently addressed the points of critique from this reviewer, except for one point which relates to further updating the methods section on MS based metabolomics. In addition, to take away reviewer concerns of the validity of methodology used for untargeted metabolomics, I would suggest that the authors provide raw data (MS peaks) to the reviewer to show how IPA (and perhaps some other relevant metabolites) was identified and quantified by untargeted metabolomics.

We thank the arbitrating reviewer for assessing our work and their feedback. We have now completed the method part as follow:

The absolute serum concentrations of 3-indolelactic acid(m/z 204 \rightarrow 142, RT=2.6), 3-indolepropionic acid (m/z 188 \rightarrow 59, RT=3.5), and tryptophan (m/z 205 \rightarrow 188, RT=0.93)) were quantitated at Metabolon facility by LC-MS/MS. Stock and internal standard solutions were prepared by dissolving an appropriate amount of material in ACN: water (1:1). These stock solutions were diluted to the Standard H level and further diluted to each of the other concentration levels. For sample preparation, serum was thawed on ice and vortexed for 10–15 s before aliquoting. For blank, blank-IS, and calibration standards, 0.0500 mL PBS was pipetted into the appropriate wells of a 2 mL 96-well plate. For matrix QC samples, 0.0500 mL QC serum was added, and for study samples, 0.0500 mL serum was dispensed into designated wells. Calibration standards were prepared by adding 0.0200 mL of the corresponding calibration spiking solution. Internal standard working solution (0.0200 mL) was added to calibration standards, blank-IS, QC samples, and study samples. Blank samples received 0.0400 mL ACN/water (1:1), while blank-IS, QC samples, and study samples received 0.0200 mL ACN/water (1:1). Methanol (0.200 mL) was then added to all wells. Plates were capped, vortexed for 1 min, and centrifuged for 10 min at 4000 rpm. A 0.100 mL aliquot of the resulting supernatant was transferred to a fresh 650 μ L 96-well plate, capped, and submitted for LC-MS/MS analysis. SAMPLE ANALYSIS: A 2.0 μ L aliquot of supernatant was injected onto an Agilent 1290/Sciex QTrap 5500 LC-MS/MS system equipped with a Waters Acquity BEH C18 column (1.7 μ m, 2.1 \times 100 mm). Chromatographic conditions were as follows: Column temperature: 50 $^{\circ}$ C, Autosampler temperature: 4 $^{\circ}$ C, Mobile Phase A1: 0.05% formic acid in water, Mobile Phase B1: 0.05% formic acid in acetonitrile, Flow rate: 0.550 mL/min, Injection volume: 2.0 μ L (adjusted as needed for sensitivity), Needle wash:

methanol/water (1:1). The MS/MS parameters were as follow Curtain Gas: 35, Collision Gas: Medium, IonSpray Voltage: -4500 V, Temperature: 500 °C, Gas 1: 70, Gas 2: 70, Scan Mode: Multiple Reaction Monitoring (MRM), negative mode. Further information can be found in Supplementary Data 6.

The peak area of the individual analyte product ions was measured against the peak area of the product ions of the corresponding internal standards. Quantitation was performed using a weighted linear least squares regression generated from fortified calibration standards prepared immediately prior to each run. The LC-MS/MS raw data is collected and processed using SCIEX software Analyst 1.6.3 and processed using SCIEX OS-MQ software.

Alternatively, targeted analysis of selected serum samples was performed at the metabolomic facility of the University of Lausanne. Serum (20 µL) was extracted with ice-cold methanol (180 µL) spiked with internal standards (IS). After centrifugation (4 °C, 21,000 g, 15 min), the supernatants were evaporated to dryness, reconstituted in water (50 µL), vortexed, and sonicated for 3 min. Each extract was then centrifuged again (4 °C, 21,000 g, 15 min) and the supernatant transferred to an LC-MS vial for LC-MS/MS analysis as described below. Calibrators were generated following the same procedure, by the addition of methanol spiked with IS mixture to each calibrator (20 µL), vortexed and transferred to LC-MS vials for the injection.

Extracts were analyzed by Reversed Phase Liquid Chromatography coupled to tandem mass spectrometry (RPLC-MS/MS) on a 6495 triple quadrupole system (QqQ) interfaced with 1290 UHPLC system (Agilent Technologies), in the analytical conditions modified from van der Velpen et al.⁹⁵. Data were acquired in Dynamic Multiple Reaction Monitoring (dMRM) mode with a total cycle time of 400 ms, applying optimized collision energies for each metabolite, using MassHunter (Agilent technologies, version B.07.00). Chromatographic separation was performed on a Acquity HSS T3 (1.8 µm 2.1 mm x 100 mm) column (Waters, Ireland), in positive ionization mode. The mobile phase consisted of A = 5 mM ammonium formate and 0.1% formic acid in H₂O and B = 100% methanol. The gradient was as follows: 0 min 0 %B, 2 min 0 %B, 4 min 5 %B, 5 min 10 %B, 7 min 90 %B, 8 min 0 %B, 11 min 0 %B. The flow rate was 300 µL/min, the column temperature 20 °C, and the injection volume 2 µL. ESI source parameters were as follows: dry gas temperature 290 °C, dry gas flow 14 L/min, nebulizer pressure 45 psi, sheath gas temperature 350 °C, sheath gas flow 12 L/min, nozzle voltage 500 V, and capillary voltage +4000 V. Further information can be found in Supplementary Data 6.

Raw LC-MS/MS data were processed using MassHunter Quantitative Analysis 10.0. Absolute quantification of indole metabolites was performed based on calibration curves and the stable isotope-labeled internal standards (IS) which were used to determine the response factor for each metabolite. Linearity of the standard curves was evaluated for each metabolite using a ten-point range; in addition, peak area integration was manually curated and corrected where necessary.

Additionally, we have further described how the identity of IPA was validated as stated in reviewer 4 response:

In untargeted metabolomics, quantification is based on MS₁ scans – and FIA is by no mean inferior to (untargeted) LC-MS. The CVs are frequently better than with LC-MS (typically <10% across the dynamic range) because they don't suffer from the errors associated with integrating 2D-chromatograms, that in metabolomics often feature asymmetric shapes and tails. Yes, FIA doesn't allow resolving isomers, but this is a quest that can be addressed ad-hoc using the proper approach and the proper samples, which is what we did in the case for 3-IPA.

Specifically, we indeed analyzed serum samples using an mixed-mode LC (<https://chemrxiv.org/engage/chemrxiv/article-details/6824924be561f77ed4c75578>) using an Agilent 6546 QTOF. At the expected m/z of 188.0717, we observed only a single peak at low

intensity in negative mode, and nothing in positive mode. Because of the low abundance, we struggled to collect an informative MS2 spectrum for spectral matching.

Therefore, we moved on to confirm the identity by retention time. According to HMDB, there are 6 isomers with formula $C_{11}H_{11}NO_2$: indole-3-propionic acid, 2-(1-methyl-1H-indol-3-yl)acetic acid, two indole variants with esterified tails, and two pyrroline derivatives with virtually no anionic group. Because of the preference in ionization in negative mode (over positive), we suspected that the peak could only be indole-3-propionic acid or 2-(1-methyl-1H-indol-3-yl)acetic acid, as all of the remaining candidates didn't have an ionizable carboxylic group. Based on these notions, we opted to use the aforementioned mixed-mode LC system because the stationary phase acts a weak anion exchanger that retains acids, but not amines (see paper). When injecting pure standards of the two candidates, we observed that indole-3-propionic acid had exactly the same retention time observed in the serum samples (< 1 sec deviation), while the other candidate eluted later and was separated at baseline. Note that all other isomers without carboxylic group elute much earlier. As the only HMDB candidate that elutes at the observed time is indole-3-acetic acid, we could confirm its identity regardless of MS2 fragmentation.

The identification of IPA was further consolidated by the targeted analyses performed in the following by Metabolon, using a different method that also matched RT and MRM and, additionally, by the University of Lausanne metabolomics facility.

Most importantly, the role and therefore the identity of IPA was validated in the functional tests that are included in the paper and demonstrate the effect IPA in potentiating the chemotherapeutic effect. As the functional validation is way more compelling than any spectral or analytical evidence, we decided to omit this long explanation from the manuscript, and included the data that initially led us to "discover" IPA, that is FIA-MS.